# Sustainability and Brazilian Agricultural Production: A Bibliometric Analysis

Rafael Araujo Nacimento [1,*], Vanessa Theodoro Rezende [2], Fábio José Muneratti Ortega [1], Sylvestre Aureliano Carvalho [1,3], Marcos Silveira Buckeridge [1,4], Augusto Hauber Gameiro [2] and Francisco Palma Rennó [2]

1 Institute of Advanced Studies, Thematic Axes Program of the University of São Paulo, São Paulo 05508-220, Brazil; fabiojmortega@yahoo.com.br (F.J.M.O.); sylvestre.carvalho@usp.br (S.A.C.); msbuck@usp.br (M.S.B.)
2 Department of Animal Production and Nutrition, University of São Paulo, São Paulo 05508-220, Brazil; vanessatrezende@usp.br (V.T.R.); gameiro@usp.br (A.H.G.); francisco.renno@usp.br (F.P.R.)
3 Department of Physics, University of Coimbra, 3004-516 Coimbra, Portugal
4 Department of Botany, Institute of Biosciences, University of São Paulo, São Paulo 05508-220, Brazil
* Correspondence: rafael.nacimento@usp.br

**Abstract:** Agriculture is one of the most important industries in the world. In this context, the importance of Brazil as a strategic country to meet a range of SDG's targets linked to food security, fighting against hunger, and poverty reduction is undeniable. This study aimed to highlight the production and dissemination of scientific research developed by Brazilian institutions, and to identify prominent authors and institutions based on articles related to sustainability, agriculture, livestock, and agribusiness. A bibliometric analysis was developed based on a sample of 3139 documents published between 2000 and 2022, comprising 21,380 authors that were then analyzed using the Biblioshiny package. As result, the term "sustainability" showed growth as it branched out to semantically similar terms, such as "sustainable agriculture" and "sustainable intensification"; and "crop–livestock integration" and "agroforestry" were highlighted as important in the development of future research. The majority of documents were produced by the University of São Paulo (~33%), the State University of São Paulo (~15%), and the Federal University of Rio Grande do Sul (~11%), suggesting that their researchers could act as coordinators in future research through the formation of multi-collaborative groups to jointly lead to the participatory elaboration of public policies that promote more sustainable paths for agricultural production.

**Keywords:** agribusiness; agricultural policy; bibliometrics; sustainable agriculture; sustainable livestock; public policies

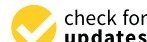



## 1. Introduction

Agriculture is one of the most important industries in the world, guaranteeing the essential food supply for more than 7.8 billion people. In this context, the importance of Brazil as a strategic country to meet a range of SDG's targets linked to food security, fighting against hunger, and poverty reduction on the planet as a whole (SDG's 1, 2, 6, 7, 12, 13) is undeniable [1]. According to data from FAO [2], in 2021, Brazil was the largest producer of soybeans (134 million tons), sugar cane (715 million tons), and coffee (2 million tons) and the third largest producer of corn (88 million tons) and meat (29 million tons), which makes it one of the main players in world agricultural production. Although it guarantees food supply, the current agricultural production model has been directly and indirectly linked to greenhouse gas emissions (climate change), loss of biodiversity, destruction of terrestrial ecosystems, excessive consumption, and water pollution, including negatively impacting five of the nine planetary boundaries proposed by Rockström et al. [3].

From the mid-20th century onwards, Brazil has been driven by technological packages linked to the Green Revolution, and the country experienced widespread growth in

agricultural production, both for export and internal use, with technological evolution in all sectors and for almost all the products. Notwithstanding, and as in other places in the world, it has been suggested that various negative externalities generated due to the expansion and intensification of agricultural production cause environmental impacts [4] and raise questions about inequalities in the distribution of government benefits [5]. Thus, the current agricultural production model has been harshly criticized [6], leading to analyses being developed to understand better the interconnections between agricultural activities, the use of environmental resources, and social aspects as a whole [7,8]. Currently, in response to the growth of socio-environmental and economic concerns, the themes "sustainability" and "agricultural production" have aroused the interest of several scientists and researchers around the world [9]. As a result, there are numerous reviews related to the topic available in the scientific literature. Such reviews involve topics such as precision agriculture [10], sustainable intensification [11], agro-ecosystems [12], use of water in agricultural production [13], biofertilizers [14], and the use of artificial intelligence in agriculture [15], among others.

More recently, using bibliometric analysis tools, Yu and Mu [16] and Sarkar et al. [9] investigated progress, themes, and their trends. They provided a comprehensive mapping of the literature correlating "sustainability" and "agricultural production." Interest in using previously published data to analyze trends from a bibliometric approach has grown in recent years [10]. According to Sarkar et al. [9], bibliometric analysis tools allowed for the following: (i) tracking the interconnection between institutions, researchers, and countries based on citation and co-citation trends; (ii) illustrating the evolution of themes; (iii) providing theoretical perspectives, compiling old and recent studies; and, consequently, (iv) revealing insights for future research. For the authors, the holistic understanding of the results obtained from bibliometric analysis tools allows for their strategic use in proposing multi-collaborative study groups by integrating the results and enabling the connection of researchers from their respective lines of research. Using the scientific knowledge generated, such groups can contribute to the discussion to propose public policies that guide agricultural production along more sustainable paths. Thus, the integration between scientific knowledge and other forms of knowledge [17] can contribute to formulating public policies for biodiversity, ecosystem services, and sustainability in agricultural production as a whole.

Given the above, this study aimed to highlight the production and dissemination of scientific research developed by Brazilian institutions and identify prominent authors and institutions based on studies on sustainability, agriculture, livestock, and agribusiness. Also, based on the collection and evaluation of articles, this study aimed to integrate significant elements to propose constructing a Brazilian research collaboration network. Such a collaborative effort can generate results and suggestions for more sustainable paths for agricultural production based on the participatory elaboration of public policies. In essence, the following research questions (RQ) were addressed:

RQ1: Which Brazilian institutions and authors are at the frontier of knowledge when correlating sustainability and agriculture, agribusiness, and livestock in research publications?

1.   RQ1-a. How many articles relating to agriculture and sustainability were published between 2000 and 2022?
2.   RQ1-b. Who are the authors, and what are the most prominent institutions of origin on the topic?
3.   RQ1-c. What are the most frequent collaborations between authors and institutions?

RQ2: What are Brazil's current themes and frontiers related to sustainability and agriculture, agribusiness, and livestock?

1.   RQ2-a. What topics (trending topics, keywords, and themes) are associated with this search topic?
2.   RQ2-b. What are the most frequent and current thematic relationships associated with sustainability and agriculture?

## 2. Materials and Methods

### 2.1. Bibliometric Analysis

Bibliometric analysis is a widely utilized method for quantitatively evaluating scientific output and trends within a particular research field. Key aspects of bibliometric analysis include examining publication patterns, citation networks, collaboration structures, and identifying influential works, authors, or institutions for the development of academic studies correlating different areas of knowledge [13]. This method offers several advantages, such as providing a systematic and objective way to assess the impact of research, uncovering emerging trends, and aiding in decision-making for future research directions [18]. Additionally, bibliometric techniques contribute to the transparency and reproducibility of the review process. However, like any methodology, bibliometric analysis comes with its set of limitations. One notable limitation is its reliance on scientific databases, which may not be exclusively designed for bibliometric purposes and can introduce errors that require careful handling [18]. The quantitative nature of bibliometrics can also pose challenges when transitioning to qualitative insights, making it crucial to supplement findings with content analysis for a more comprehensive understanding. Furthermore, while bibliometric studies offer valuable short-term insights, making ambitious claims about the long-term impact of a research field may be challenging [19]. Despite these limitations, bibliometric analysis remains a valuable tool for navigating the complex landscape of scientific knowledge.

This study used bibliometric analysis to identify trends and key factors based on published works, the main authors, and research institutions, with sustainability in agriculture as the theme.

### 2.2. Defining Objectives

Bibliometric analysis was used to highlight the production and dissemination of scientific studies produced by Brazilian institutions and identify authors and institutions to propose the formation of a collaborative network that directs agricultural national output towards more sustainable paths through the elaboration of participatory public policies. Analyzes were developed to observe the trends in the thematic evolution of publications and the main authors. To this end, data relating to (a) number of publications, (b) authors, and (c) author keywords and keywords Plus were obtained from the scientific literature. For authors, indicators related to the number of publications over the years reflect interest in a given topic and the number of publications over the years per author to reflect their performance in the area of knowledge.

Keywords Plus are words or phrases that appear frequently in the titles of an article's references but not in the article's title. Keywords Plus increases the power of cited reference searching, searching across disciplines for all articles that have cited references in common [20]. They can be used to decipher key terms and compare their origins and are vital parameters for extracting content and scientific concepts expressed in articles [21]. On the other hand, according to Abafe et al. [22], using author keywords allows for a better inductive analysis of the content presented in documents than keywords Plus. For the authors, indicators related to the frequency of use of keywords indicate themes of interest and make it possible to infer the trend in thematic evolution.

### 2.3. Data Collection

The sample used was obtained from the Web of Science (WoS) database. WoS is justified because it covers more than 20,300 scientific journals, books, and conferences with more than 71 million research materials [23] and has greater information coverage [9]. Also, the choice to use Web of Science as our primary database was influenced by several factors. Web of Science is a widely recognized and extensively used database, particularly in the academic and research community. It is known for its comprehensive coverage of peer-reviewed journals across various disciplines. The scope of this research, focusing on sustainability, agriculture, and related fields, aligns well with the content indexed in

Web of Science, showing a robust platform for bibliometric analysis. While Scopus and Google Scholar are valuable resources, Web of Science has been integrated into the Scielo platform since 2014, where a significant portion of peer-reviewed journals in multiple languages is indexed [24,25]. Moreover, a recent study showed that despite evidence that Google Scholar significantly retrieves more citations than the WoS Core Collection and Scopus in all thematic areas, all citations found by WoS (95%) and Scopus (92%) were also found by Google Scholar [26]. However, not all documents indexed by Google Scholar undergo a rigorous peer review process. For these reasons, we chose the Web of Science database to guide our bibliometric analysis. In this sense, documents published in other languages, with titles, abstracts, and keywords in English, were included. Thus, the search covered documents published in English, Portuguese, French, and Spanish. The Boolean search consisted of retrieving peer-reviewed scientific articles written in English between 2000 and 2022. The retrieval of scientific articles is justified because they have greater academic value, as suggested by Freire and Nicol [27]. To respond to the research objectives, only Brazilian studies were considered based on a selection using country filtering. The search included the following query: (TITLE-ABS-KEY (sustain* AND agriculture); TITLE-ABS-KEY(sustain* AND crop); TITLE-ABS-KEY(sustain* AND agribusiness); AND TITLE-ABS-KEY (sustain* AND livestock); AND TITLE-ABS-KEY (sustain* AND poultry); AND TITLE-ABS-KEY (sustain* AND broiler); AND TITLE-ABS-KEY (sustain* AND dairy); AND TITLE-ABS -KEY (sustain* AND pig); AND TITLE-ABS-KEY (sustain* AND swine). The Boolean operator "asterisk" (*) was used to the right of the word "sustain" to ensure that the words "sustainability" and "sustainable" were included.

### 2.4. Statistical Analysis and Tools

The sample metadata were incorporated into Biblioshiny, an interface for the Bibliometrix package used by RStudio [28,29]. This method adds an analytical dimension to the study of the selected sample, aiming to achieve more meticulous and insightful results. As highlighted by Donthu et al. [18], bibliometric analysis offers advantages that surpass the limitations of relying solely on literature reviews as the primary analysis method. This is attributed to the broader scope of bibliometric studies, which embrace a more quantitative nature, reducing the likelihood of biased study results due to a narrow research focus. In addition to Biblioshiny, we integrated VOSviewer and Mathematica into our analysis, enhancing the visualization and validation of our results. VOSviewer, renowned for its network analysis capabilities, allowed us to delve into collaboration patterns and thematic clusters within the literature. Mathematica, with its versatile numerical tools, complemented our approach by providing additional insights and facilitating a comprehensive examination of the bibliometric landscape. Together, these software tools were pivotal components of our analysis, ensuring a robust and multifaceted exploration of the bibliometric data.

The methods employed comprised performance analysis and mapping science. For performance analysis, publications were examined concerning authorship and research institutions, while mapping science identified trends in scientific research based on bibliometric tools intrinsic to the program used. Based on the relationship between time, documents produced, and citation per document, the h-index, g-index, and m-index were used to highlight the most prominent authors. The h-index is determined by sorting the author's publications in descending order of the number of citations each has received. The h-index is the highest number where the author has many publications with at least that many citations each. Equation: If the author has "h" papers that have at least "h" citations each, but not more than "h + 1" citations, then their h-index is "h." The m-index is another variant considering the number of publications and their citations. It is calculated by dividing the sum of the square roots of the number of citations of each paper by the square root of the total number of publications. Equation: $M = \Sigma\sqrt{c}/\sqrt{N}$, where $\Sigma\sqrt{c}$ is the sum of the square roots of the number of citations, and $\sqrt{N}$ is the square root of the total number of publications. The g-index considers the number of citations and the "weight" of

each publication. It is calculated by dividing the sum of the square roots of the number of citations of each paper by the sum of the square roots of the natural numbers. Equation: $G = (\Sigma\sqrt{c})/(\Sigma\sqrt{n})$, where $\Sigma\sqrt{c}$ is the sum of the square roots of the number of citations, and $\Sigma\sqrt{n}$ is the sum of the square roots of the natural numbers (publication order) [30,31].

According to Valesco-Muñoz et al. [13], the combination of techniques gives the literature evaluations a quantitative rigor for previously subjective analyses and provides evidence of theoretically defined categories in article reviews. Bibliometric mapping was developed for mapping science based on the 7937 authors' keywords for indexing their documents and the 6579 words provided by the WoS database (keywords Plus). Keyword dynamics and trending topics were analyzed using author keywords and keywords Plus. The author's keywords were used to investigate (a) the thematic evolution of the keywords, (b) frequency of occurrence, density, and analysis of co-occurrence networks, (c) the development of thematic maps, and (d) trending topics. Keywords Plus was used to develop (a) the tree map and (b) grouping by joining documents. Also, both keywords were used to investigate the interrelationship between institutions and authors. To develop the thematic map, the analyzed time window was divided into five parts (2000–2010, 2011–2015, 2016–2020, 2021, and 2022). The leading eigenvalues algorithm was used in all analyses involving clusters (co-occurrence analysis in networks, three-field graph, clustering by union, thematic maps, and thematic evolution). The other parameters involved in the analysis followed the standard Biblioshiny configuration.

## 3. Results and Discussion

### 3.1. Descriptive Analysis of the Selected Sample

Following the methodology, the study identified 3139 documents, comprising 21,380 authors distributed among all documents (Figure 1b). The annual publication rate of documents was ~143, presenting a yearly growth rate of ~17%. The average age of documents is ~6 years, indicating that most documents were published in the last 6 years. It is noted that interest in the topic studied gained prominence from 2007 onwards, with exponential growth since then, with its peak being observed in 2021 (n = 546 publications), representing ~17% of the sample.

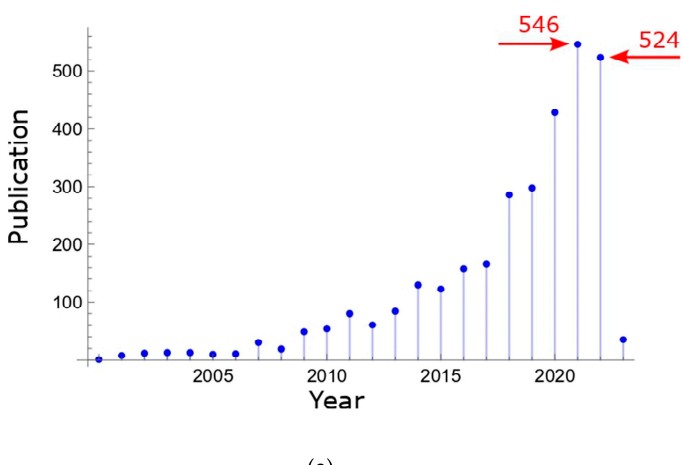

| Description | WOS [†] |
|---|---|
| Base Information | |
| Sources, (no) | 753 |
| Documents, (no) | 3139 |
| Annual growth rate, (%) | 16.72 |
| Document average age | 6.01 |
| No. of citations/document | 21.32 |
| References, (no.) | 122,759 |
| Documents | |
| Keywords Plus (ID), (no.) | 6579 |
| Author's keywords (DE), (no.) | 7937 |
| Authors | |
| Authors, (no.) | 21,380 |
| Single author documents, (no.) | 37 |
| Collaboration | |
| Single author documents, (no.) | 42 |
| Co-authorship/document | 11.3 |
| International co-authorship, (%) | 48.3 |

(**a**)  (**b**)

**Figure 1.** Statistical summary of bibliometric metadata for publications of scientific articles obtained from the Web of Science (WoS). (**a**) Distribution of the number of publications of Brazilian scientific works per year; (**b**) summary of statistics for the sample collected.Note: (**a**) *x*-axis: years; y axis: published documents; (**b**) Sources—newspapers, books, etc.; keywords Plus (Keywords Plus (ID)); author's keywords (DE)); † articles: 3051; data paper: 7; prior access: 12; proceedings paper: 69.

In this study, it is possible to suggest the growth of interest in research related to the topic following the boom in publications in 2007, since ~97% of documents were produced between 2007 and 2022. The growth of publications correlating sustainability and agricultural production in this period is corroborated by the progression of research production related to sustainable agriculture found by Sarkar et al. [9]. For Sarkar et al. [9], such growth may result from the prominence of sustainability in contemporary agro-economic perspectives. For Yu and Mu [16], sustainable development has been elevated to a national strategy and has migrated from the theoretical field of discussion to the practical field of implementation. As a result, its implementation and promotion of strategies for sustainable development have directly impacted the number of publications year after year. As an example, using the study by Abafe et al. [22], the author observed that, in recent years, aspects that correlate with the sustainable use of water in agriculture have become a severe concern of society, causing more research and strategies to be implemented for the rational use of the resource.

Consequently, it is possible to suggest that interest in more general aspects of sustainability and agricultural production follows the same trend, including in Brazil. In turn, the increase in interest positively impacts the creation of global research fields of interest to scientists and affected communities, which increases the number of publications on the topic. Furthermore, it is possible to observe the alignment of interest in the subject of study between Brazilian and foreign researchers since the interest boom occurs at similar times.

### 3.2. Identification of Institutions, Co-Authorships, and Collaborations

The results indicate that the University of São Paulo (USP, n = 1043), São Paulo State University of São Paulo (UNESP, n = 670), Federal University of Rio Grande do Sul (UFRGS, n = 349), and the Federal University of Viçosa (UFV, n = 262) have the highest numerical production of studies (Figure 2a). The four institutions represent ~67% of all document publications sampled. It is also possible to highlight the centrality of USP as well as its collaboration networks between national institutions (e.g., Universidade Estadual Paulista and Universidade Federal de Minas Gerais) and international (e.g., Ghent University, Belgium; and Universidad Nacional Autónoma de México, Mexico) (Figure 2b).

The prominence of the institutions listed in this study can be attributed to two factors: (i) the number of research programs with high-quality scores and (ii) the available financial support. Regarding the scores ranging, the research programs associated with Brazilian institutions periodically undergo governmental quality evaluations. In these evaluations, five criteria are considered: program proposal, teaching staff, student body, theses and dissertations, intellectual production, and social insertion. Ultimately, these programs receive scores ranging from 1 to 7. A score of 5 indicates that the course is considered 'very good', while scores of 6 and 7 are awarded to programs of excellence at the national and international levels, respectively. Among the 190 Brazilian research programs linked to the agricultural sciences field with scores between 5 and 7, approximately 57% (108 research programs) are affiliated with the top 10 institutions. Within this group, about 12% (23 research programs) are associated with USP, which boasts the highest number of research programs with international excellence (seven research programs). Most of these programs have over 50 years of experience, suggesting a high level of maturity (these data were obtained using public consultation in the Coordination of Superior Level Staff Improvement (Capes). Available at: https://sucupira-v2.capes.gov.br/sucupira4/programas?grande-area-conhecimento=5&search=&size=100&page=4 (accessed on 25 January 2024).

On the other hand, the financial support is closely related to these scores and their maintenance, for instance, specific financial resource programs to support academic excellence programs (Programa de Excelência Acadêmica (Proex/Capes/MEC)). According to the National Council for Scientific and Technological Development database (CNPq) (these data were obtained using public consultation to the website of CNPq Data Transparency (https://portaldatransparencia.gov.br/ (accessed on 25 January 2024)), other crucial federal research support agencies, and approximately 1350 research institutions received financial

backing for research development between 2002 and 2022 (including research grants, financial project support, etc.). The top 10 institutions garnered 48% (BRL 1.7 billion) of the total financial resources available. UFV, USP, UNESP, and UFRGS collectively received 26% (BRL 941.5 million) of the overall financial resources. Notably, this figure pertains specifically to the financial support allocated for the agricultural sciences field. Thus, it is possible to suggest that the prominence of those institutions has been related to a higher number of research programs that present high-quality score ranging.

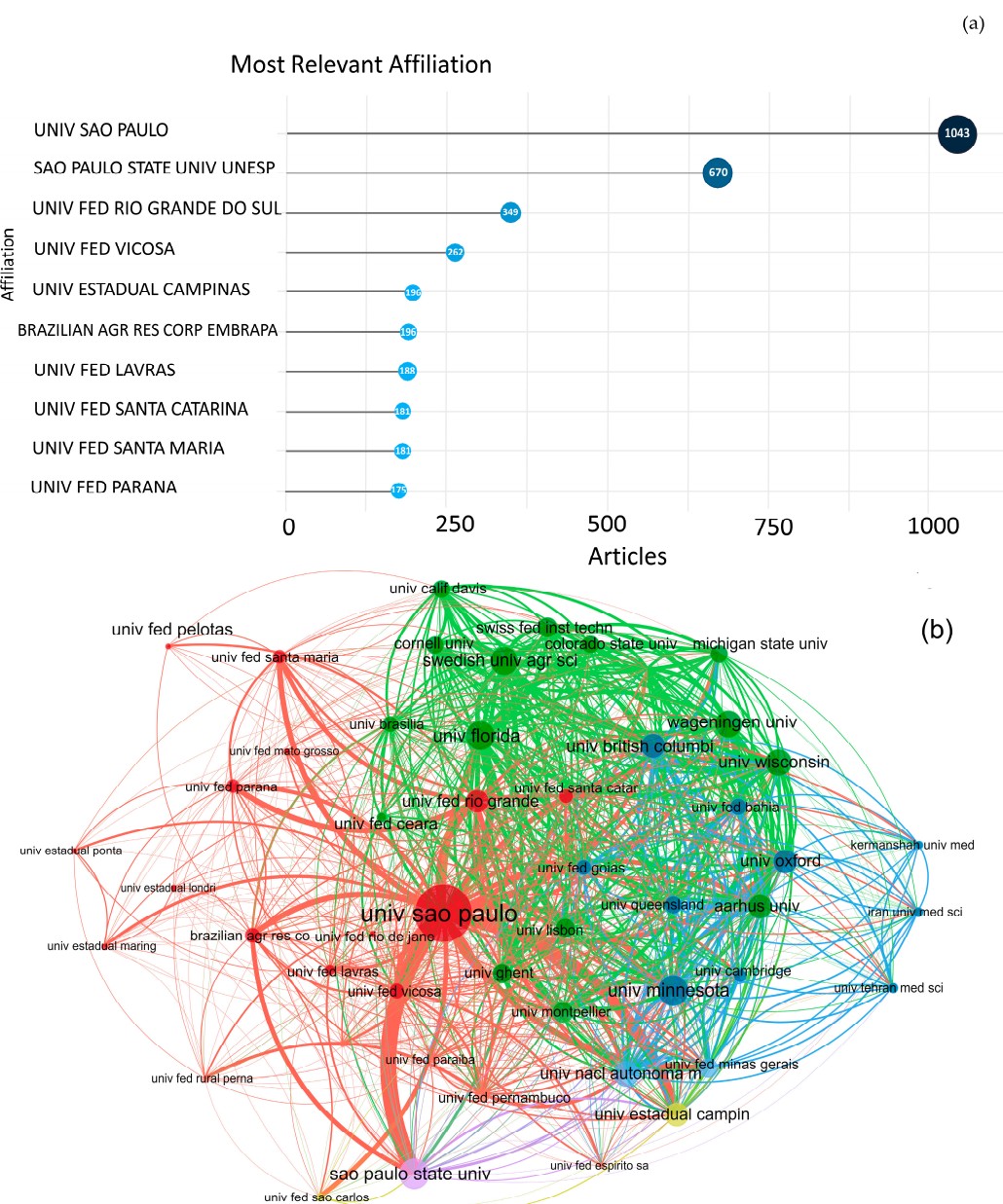

**Figure 2.** Distribution of papers by Brazilian institutions and collaboration networks of affiliations using bibliometric metadata from Web of Science (WoS). (**a**) Top 10 affiliations in article count; (**b**) collaboration network among affiliations. Nodes represent institutions, and links depict collaboration between two affiliations. Due to its many published articles, the University of São Paulo is highlighted as a major node. Nodes of size proportional to the number of documents published represent other institutions. Thicker links indicate the high intensity of collaboration between the institutions concerning the publication of papers. The color of an item is determined by the cluster to which the item belongs.

Figure 3 presents the authors' co-occurrence network analysis. For Velasco-Muñoz et al. [13], the co-occurrence network analysis explores the social networks researchers form when collaborating on publications. In this way, connections between authors are formed when they produce a document jointly. Consequently, collaboration between authors can form a cooperative relationship between institutions [16].

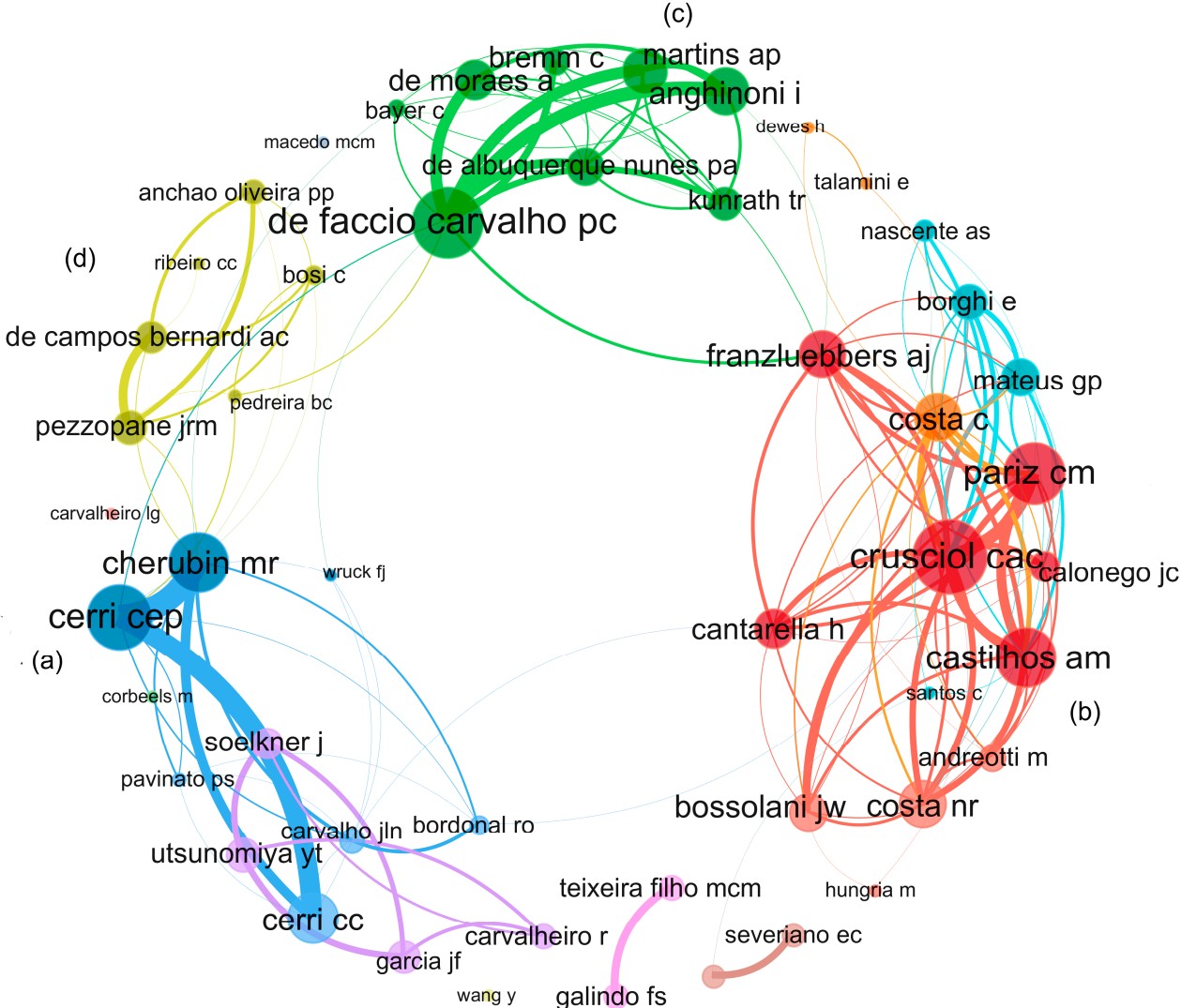

**Figure 3.** Author's co-occurrence network. Nodes represent authors, and links indicate collaboration between them for document production. Larger nodes represent authors with a greater number of publications. The thickness of the links connecting authors signifies the intensity of collaboration, with thicker lines indicating a higher degree of cooperation between authors. The color of each node corresponds to the cluster to which the author belongs. Affiliation clusters are identified as follows: (**a**) University of São Paulo (USP—blue); (**b**) São Paulo State University (UNESP—red); (**c**) Federal University of Rio Grande do Sul (UFRGS—green); and (**d**) Brazilian Agricultural Research Corporation (Embrapa—gold).

It is observed that four researcher groups presented a collaboration network with more than four researchers. Among them, the groups formed by authors from USP (a) in purple), UNESP (b), UFRGS (c), and Embrapa (d) stand out. It can be observed that, although collaboration between researchers is more internally concentrated, there are collaborations between institutions, identified based on the flows in gray lines that depart from one group to another. It is worth mentioning that flows with greater thickness represent a greater cooperation density between authors. This finding indicates that, jointly, USP, UFRGS, and

UNESP, can substantially influence the direction of future studies related to sustainability in agriculture based on multi-collaborative research, both between these institutions and among the others listed, promoting specialization, division of work [32], and promoting the dissemination of local studies to the scientific community as a whole.

### 3.3. Authors

The analysis of the authors' activity is demonstrated in Figure 4. The figures comprised the 20 authors with the highest numerical productivity over time based on the h-, g-, and m-indexes.

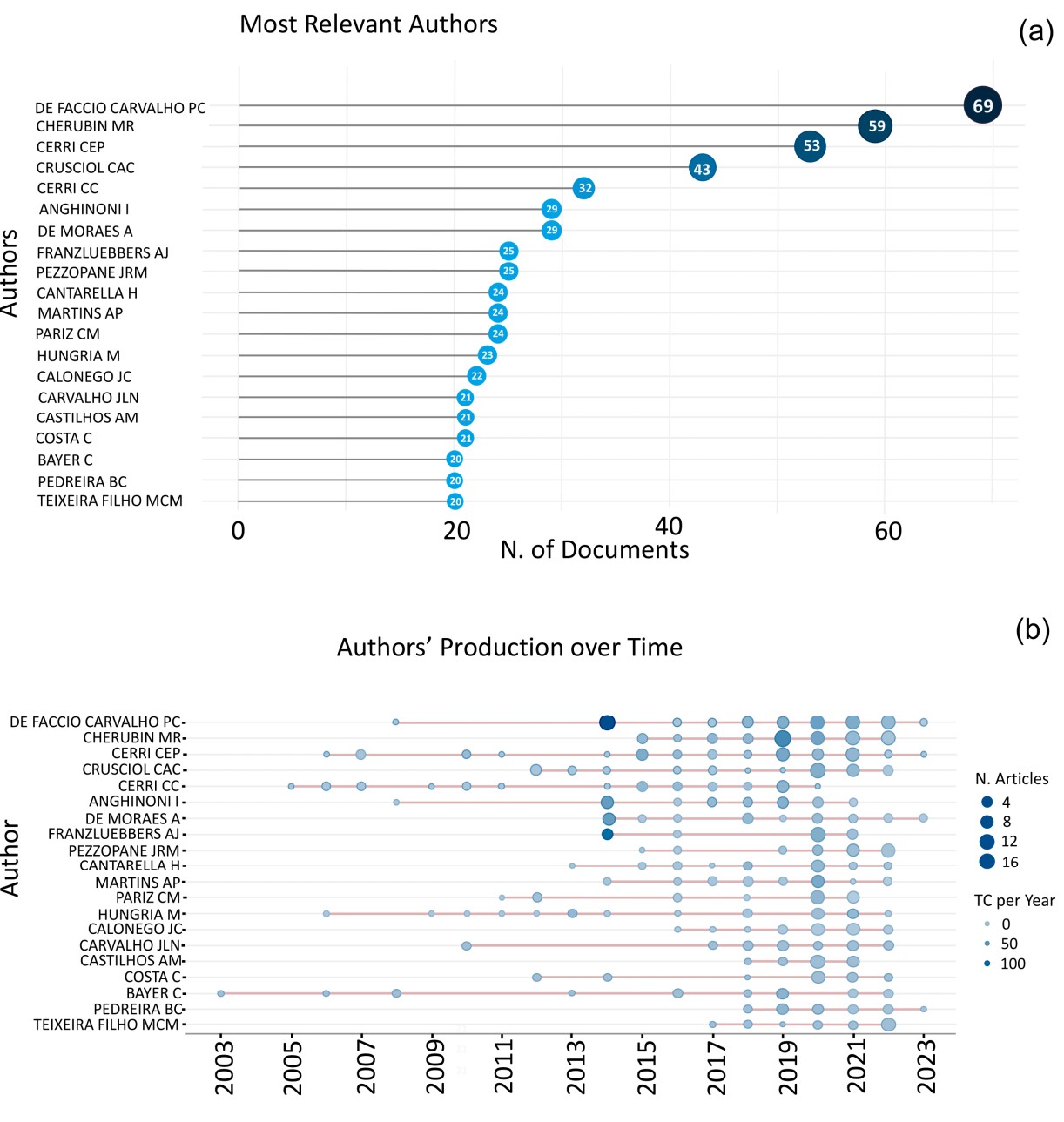

**Figure 4.** Analysis of authors' activity over the years. (**a**) Selection of the 20 most productive authors according to the number of documents produced; and (**b**) number of articles and total citations (TC) over time (2000–2022).

Starting from the number of documents produced per author, the results suggest Faccio Carvalho, P. C.; Cherubin, M. R.; Cerri, C. E. P; Crusciol C. A. C.; and Cerri, C. C. as the five most numerically productive authors. Faccio Carvalho, P. C., has been a professor at the Department of Forage Plants and Agrometeorology at UFRGS since 1997, focusing on research and extension in the soil–plant–animal relationship. His studies encompass natural environments, cultivated lands, and integrated pasture grazing [33]. Cherubin, M. R., is a professor at the Department of Soil Science at ESALQ-USP. His studies center around quantifying and comprehending the impacts of land use, emphasizing the management and health of soil practices, carbon dynamics, and the provision of ecosystem services in both natural and agricultural ecosystems [33]. Cerri, C. E. P., is a professor at the Department of Soil Science at ESALQ-USP, with a focus on themes related to soil organic matter, global warming, climate changes, agriculture, and carbon credits, mathematical models, geostatistics, and geoprocessing [33]. Cerri, C. C., has been a professor at the Center of Nuclear Energy in Agriculture (CENA/USP) at USP since 1975. He specializes in researching soil organic matter dynamics in natural ecosystems and those modified by agricultural practices, livestock, and reforestation. His studies address greenhouse gas emissions, climate changes, and land-use changes associated with agriculture and livestock production practices [33]. Crusciol, C. A. C., is a professor at the Faculty of Agricultural Science at UNESP (FCA/UNESP). His expertise is concentrated on agricultural production systems, soil fertility management, plant nutrition, and applied vegetal physiology [33].

Faccio Carvalho, P. C.; Cherubin, M. R.; and Cerri, C. E. P. were responsible for ~2% of the total number of published studies (Figure 4a), indicating heterogeneity in document production. The five authors were responsible for publishing ~8% of the total sample. It is worth highlighting that both the production and the number of citations of Faccio Carvalho, P. C. and Cherubin, M. R. showed growth after 2013 (Figure 4b). However, although Faccio Carvalho, P. C. has presented studies since 2008, authors such as Bayer, C.; Cerri, C. E. P.; Cerri, C. C., and Hungary, M. shown oldest contributions, with publications before 2007. Like Faccio Carvalho, P. C., these authors presented a long sequence of contributions relating to sustainability and agricultural production. On the other hand, the results pointed to the concentration of publications and citations from 2015 onwards, which leads to the inference of an increase in the number of eminent researchers dedicated to studying sustainability in agriculture. With this, it is possible to suggest the importance and continuity of the area of knowledge.

According to the h-index, Faccio Carvalho, P. C. and Cherubin, M. R. present the highest value for the studied sample (n = 24), followed by Cerri, C. C. and Cerri, C. E. P., both with 22 (Figure 5a). The h-index is an index that attempts to level authors by calibrating their production to the impact of citations of their publications. The index comprises a set of the researcher's most frequently referenced publications and how often those works were cited in other studies [22]. Considering the Faccio Carvalho, P. C. and Cherubin, M. R. results, although the h-index tends to increase with career time, the difference in the authors' document production longevity was not a sufficient weight to distinguish them [34].

Notably, when considering the g-index, Faccio Carvalho, P. C. presents the best result (n = 46), followed by Cerri, C. E. P. (n = 38), with Cherubin, M. R. being third (Figure 5b). The g-index is a variant of the h-index. It seeks to emphasize densely cited studies, providing a greater impact of this study on the sample selected based on its citations within the time window. In the g-index, longer-lived authors present better results due to their longer careers, which may increase the number of citations and the relevance of their publications [35].

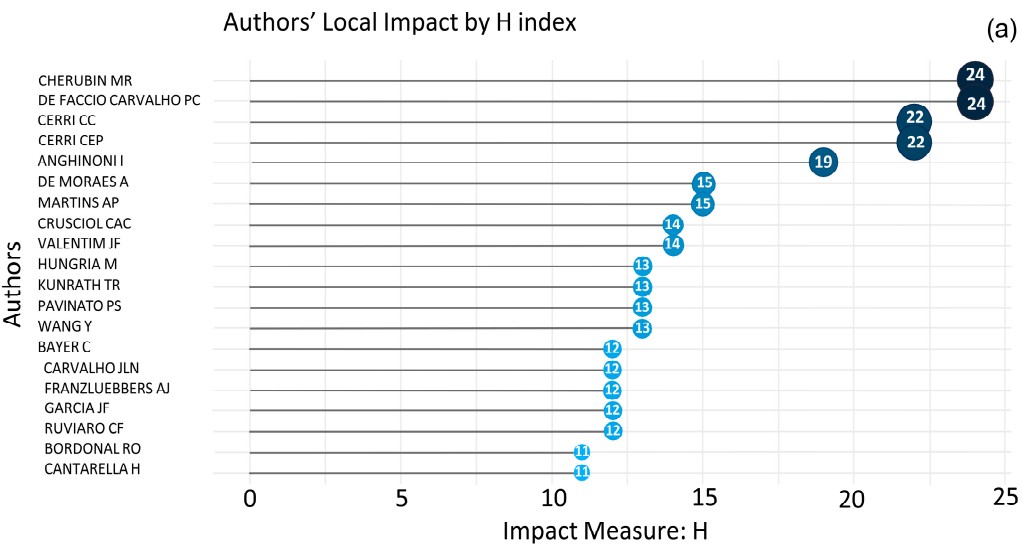

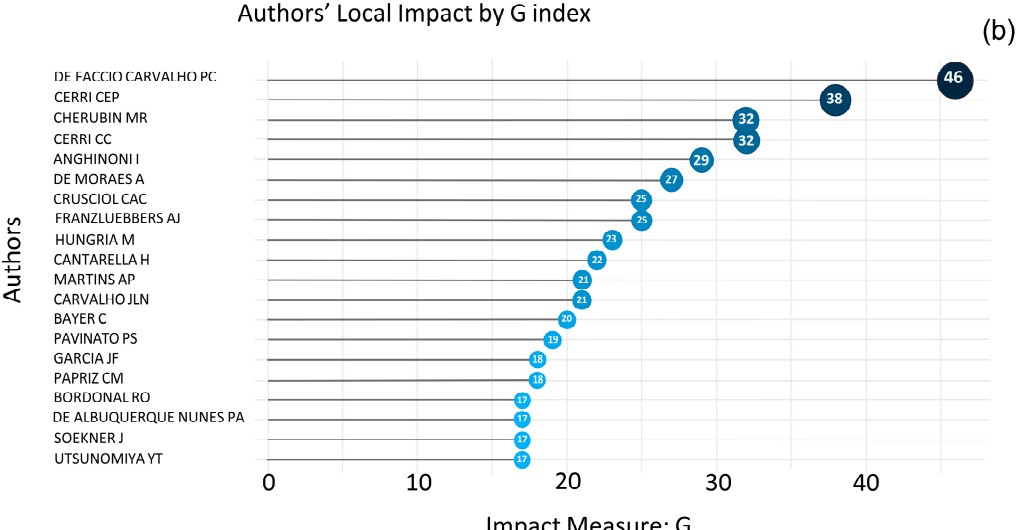

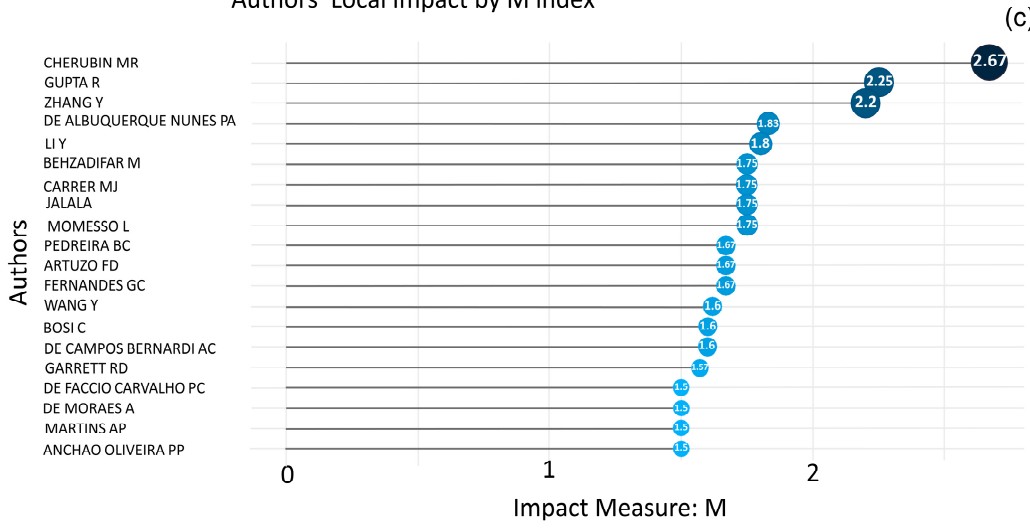

**Figure 5.** Analysis of the author's activity based on h-index (**a**), g-index (**b**), and m-index (**c**).

On the other hand, considering the m-index, Cherubin, M. R. presents the highest result (2.67), suggesting that this is a prosperous author in the study area (Figure 5c). The m-index is another variant of the h-index. Using years as a weight, this index considers the volume of works produced yearly since the author's first publication and compared to authors such as Faccio Carvalho, P. C., Cerri, C. E. P., and Cerri, C. C., Cherubin, M. R. had a shorter contribution time, which infers a shorter career time in research development and can make the comparison among the authors in the sample unfair. However, since the m-index provides a fairer comparison between authors with different career times [36], it is possible to support the suggestion that Cherubin, M. R. is one of the most promising authors in the study area. Thus, authors with good numerical productivity at the beginning of their careers tend to present better results for the m-index.

It is important to highlight that the list of authors presented in the study was necessary solely and exclusively to present the methodology used to map the authors. Such results are restricted to the search methods and, consequently, to the sample in the database consulted. Furthermore, the intention of the present study was not to rank researchers as better or worse since each author, even those not listed, knows different areas of interest but of equal need, which could contribute to greater sustainability in agricultural production. In this sense, the main contribution of the methodology was to list potential promising authors for forming public policies that aim to propose solutions to the different problems that permeate the agricultural sector "from cradle to cradle". With this, authors with longer and shorter careers could work in a multi-collaborative way to solve problems, combining practical experience accumulated in years of research with the most recently produced knowledge.

### 3.4. Citation Assessment

Regarding the citations, there are 122,759 citations and an average of 21.32 citations/document. Table 1 shows the top 10 most globally cited documents, with Lehmann et al. [37], Kattge et al. [38], and Poorter et al. [39] being the most cited documents. The top 10 most cited documents account for 5% of all citations, being published between 2003 and 2020. The document of Lehmann et al. [37] was the most prestigious among the others, presenting twice citations of the second and third most cited document. It is important to highlight that this study had the collaboration of Embrapa Amazônia Central. In general, Lehmann et al. [37] aimed: (a) to compare the fertility of an archaeological Anthrosol (i.e., Amazonian dark earth soils) to a typical upland Ferralsol of the central Amazon basin; (b) to assess how these Anthrosols maintain their high nutrient availability; and finally (c) to compare the effect of inorganic and organic amendments such as charcoal and manure on fertility and nutrient retention of a Ferralsol with those of an archaeological Anthrosol. For example, this document was used to highlight different features from anthropogenically made soils [40,41]; the application of some features observed in anthropogenically made soils as technologies (i.e., biochar fertilization) to the soil quality improvement (i.e., soil pH regulation and nutrient concentration improvement [42–44], water retention [45], etc.) and as potential tools for sustainable agriculture [44,46]. On the other hand, when considering the number of citations per year, the document of Kattge et al. [38] showed a higher number of citations. In general, this document introduces the latest version of a plant trait database. It discusses the importance of plant traits in understanding the functioning of ecosystems and the potential applications of the database in various research fields. This database provides a worldwide collection of plant trait data. Studies cited in this document highlight the initiative in data sharing and synthesis of plant trait data [47–49].

**Table 1.** Top 10 most globally cited documents.

| Paper | DOI | Total Citations | TC per Year | Normalized TC |
|---|---|---|---|---|
| LEHMANN J, 2003, PLANT SOIL | 10.1023/A:1022833116184 [37] | 1241 | 56.41 | 10.06 |
| KATTGE J, 2020, GLOB CHANGE BIOL | 10.1111/gcb.14904 [38] | 647 | 129.40 | 43.92 |
| POORTER L, 2016, NATURE | 10.1038/nature16512 [39] | 626 | 69.56 | 13.11 |
| KLEIJN D, 2015, NAT COMMUN | 10.1038/ncomms8414 [50] | 559 | 55.90 | 13.91 |
| BRUSSAARD L, 2007, AGRIC ECOSYST ENVIRON | 10.1016/j.agee.2006.12.013 [51] | 529 | 29.39 | 9.34 |
| RADER R, 2016, PROC NATL ACAD SCI U S A | 10.1073/pnas.1517092112 [52] | 475 | 52.78 | 9.95 |
| BASSU S, 2014, GLOB CHANGE BIOL | 10.1111/gcb.12520 [53] | 440 | 40.00 | 8.69 |
| PEOPLES MB, 2009, SYMBIOSIS | 10.1007/BF03179980 [54] | 439 | 27.44 | 7.71 |
| PASTORELLO G, 2020, SCI DATA | 10.1038/s41597-020-0534-3 [55] | 432 | 86.40 | 29.33 |
| VAN DEN HOOGEN J, 2019, NATURE | 10.1038/s41586-019-1418-6 [56] | 422 | 70.33 | 17.26 |

Considering the top 10 most locally cited documents (Table 2), De Moraes et al. [57] was the most influential document, showing more than twice the citations of the second and third most locally cited documents. This study presents the benefits of integrated crop–livestock systems for more sustainable agriculture production. Regarding soil–plant–animal aspects, the study highlights the greater environmental gains and less vulnerability, higher yields, and more financial gain of integrated crop–livestock systems compared to monocultures or non-integrated livestock farming. It is important to highlight the local citations, considering the number of times an author (or a piece of a document) in the sample has been referenced by other authors who are also authors in the studied sample [22].

**Table 2.** Top 10 most locally cited documents.

| Document | DOI | Year | Local Citations | Global Citations | LC/GC Ratio (%) | Normalized Local Citations | Normalized Global Citations |
|---|---|---|---|---|---|---|---|
| DE MORAES A, 2014, EUR J AGRON | 10.1016/j.eja.2013.10.004 [57] | 2014 | 87 | 159 | 54.72 | 29.00 | 3.14 |
| MACEDO MCM, 2009, REV BRAS ZOOTECN | 10.1590/S1516-35982009001300015 [58] | 2009 | 39 | 126 | 30.95 | 13.55 | 2.21 |
| CRUSCIOL CAC, 2012, AGRON J | 10.2134/agronj2012.0002 [58] | 2012 | 28 | 63 | 44.44 | 12.56 | 2.08 |
| DICK M, 2015, J CLEAN PROD-a | 10.1016/j.jclepro.2014.01.080 [59] | 2015 | 27 | 87 | 31.03 | 15.97 | 2.16 |
| HUNGRIA M, 2010, PLANT SOIL | 10.1007/s11104-009-0262-0 [60] | 2010 | 24 | 324 | 7.41 | 25.41 | 8.66 |
| COSTA MP, 2018, J CLEAN PROD | 10.1016/j.jclepro.2017.10.063 [61] | 2018 | 23 | 55 | 41.82 | 28.23 | 2.47 |
| VILELA L, 2011, PESQUI AGROPECU BRAS | 10.1590/S0100-204X2011001000003 [62] | 2011 | 22 | 63 | 34.92 | 22.28 | 2.92 |
| BIELUCZYK W, 2020, GEODERMA | 10.1016/j.geoderma.2020.114368 [63] | 2020 | 21 | 24 | 87.50 | 28.62 | 1.63 |
| RUVIARO CF, 2012, J CLEAN PROD | 10.1016/j.jclepro.2011.10.015 [64] | 2012 | 19 | 86 | 22.09 | 8.52 | 2.84 |
| FORTES C, 2012, BIOMASS BIOENERG | 10.1016/j.biombioe.2012.03.011 [65] | 2012 | 19 | 94 | 20.21 | 8.52 | 3.10 |

Meanwhile, in terms of most locally cited references (Figure 6), De Moraes et al. [22] was the third most cited, after Alvares et al. [66] and R Core Team, the first and second most cited references, respectively. These references were generally cited as tool presentations used in experimental conditions in the documents to highlight statistical procedures and describe experimental conditions.

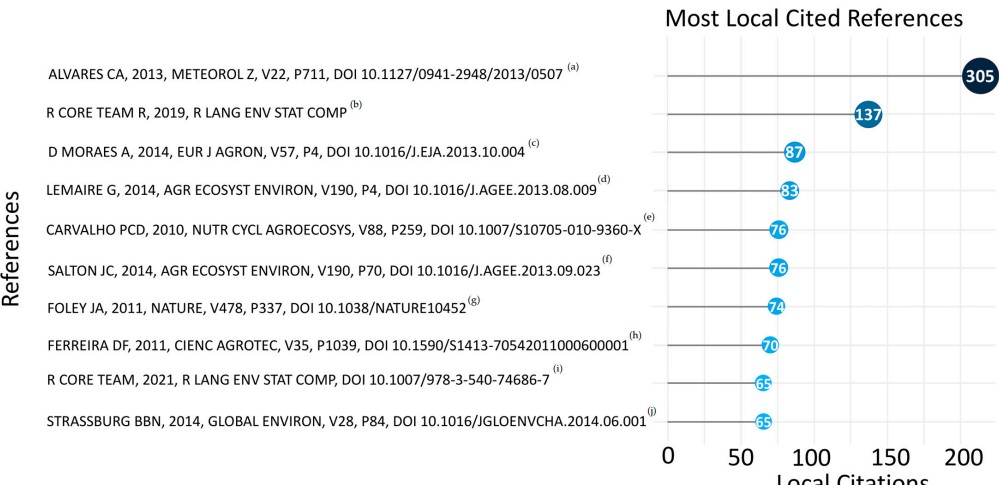

**Figure 6.** Top 10 most locally cited references. (a) is Alvares et al. [66], (b,i) is R Core Team [67], (c) is De Moraes et al. [57], (d) is Lemaire et al. [68], (e) is de Faccio Carvalho et al. [69], (f) is Salton et al. [70], (g) is Foley et al. [71], (h) is Ferreira [72], (j) is Strassburg et al. [73].

In summary, the sampled literature was mainly composed of experimental research interested in yield and production improvement (i.e., the use of biochar in vegetal production) and long-term studies perspectives interested in animal–vegetal integration. Also, the number of citations was generally high and provided different perspectives to explore sustainability in agriculture and livestock production systems.

### 3.5. Analysis of the Mapping of Keyword Co-Occurrence Networks

Keywords represent the focus of documents and their structure to a certain extent [16]. They illustrate the fundamental objective of the document and can be used to explain the core motivation of a study in detail [74]. The 10 most relevant terms occurring as the authors' keywords were: "sustainability" (305 occurrences), "agriculture" (109 occurrences), "Brazil" (107 occurrences), "sustainable agriculture" (96 occurrences), "agribusiness" (76 occurrences), "soil" (63 occurrences), "soy" (57 occurrences), "livestock" (54 occurrences), "sustainable intensification" (54 occurrences), and "management" (52 occurrences; Figure 7).

## Most Relevant Words

**Figure 7.** Top 20 of the author's keywords with the highest frequency of occurrence.

The keywords "sustainability", "sustainable", "agriculture", "agribusiness", "livestock", and "sustainable agriculture" partially mirrored the queries made in the database.

This result suggests that these terms may be the most discussed areas in Brazil's research on sustainability and agriculture [22]. Other terms such as "soil", "soybean", and "sustainable intensification", among the others listed, may have direct or indirect applications for the sustainability of agricultural systems [22]. In turn, the word cloud more visibly presents the density and frequency with which the keywords occurred. It is possible to highlight words that, although with lower frequency and density (e.g., "food security", "bioenergy", "climate change", and "ecosystem services"), emerge as potential applications in research related to the sustainability of agricultural systems (Figure 8). Furthermore, for Mulay et al. [75], words in smaller letters indicate potential study directions.

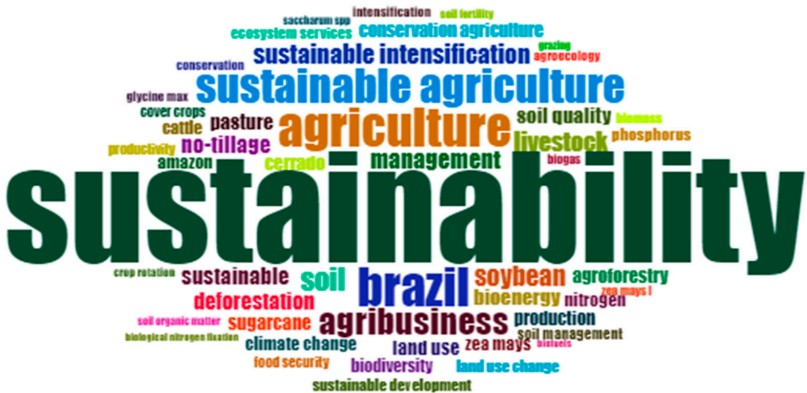

**Figure 8.** Word cloud from author keywords.

When the interrelations between the keywords are observed from the co-occurrence networks, it is noted: (i) the centrality of the term "sustainability" as well as (ii) its relationship with "agribusiness", "Brazil", and "agriculture", which allowed us to infer a possible direction for the areas of agricultural studies towards the development of research aimed at more sustainable agricultural practices (Figure 9). As an example of practices, one can infer the reduction in deforestation in the Amazon and the Cerrado regions. We also note the emergence of less frequent relationships, which are of great relevance. As an example, we can mention "food security", "climate change", and "agroecology", suggesting potential study directions for future themes.

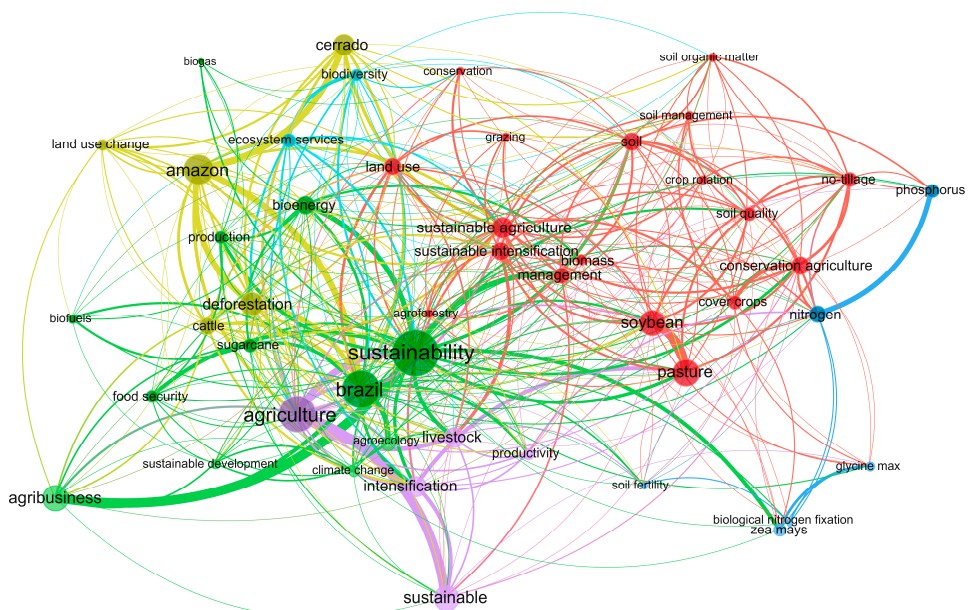

**Figure 9.** Analysis of the mapping of co-occurrence networks of the author's keywords. Thicker flows represent a more significant relationship among keywords in documents.

Regarding the tree map results, the keywords Plus "management/management" (12%), followed by "yield" (8%) and "growth" (7%) stand out. Thus, it is possible to infer that zootechnical or phytotechnical performance has directed the main focuses of the research used as a reference to guide the writing of the sampled documents (Figure 10).

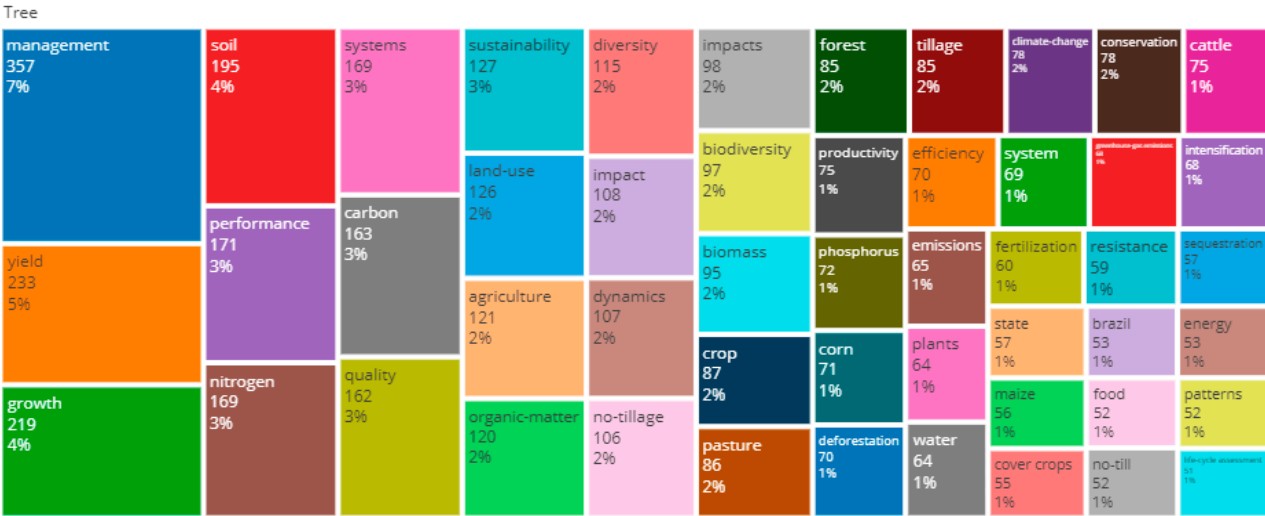

**Figure 10.** Treemap of keywords Plus showing the percentage value of keyword occurrence concerning the total.

*3.6. Suggestion for Identifying and Forming Multi-Collaborative Research Groups*

Figure 11 presents a more precise overview of how the methodology used can suggest the formation of research groups correlating institutions and authors based on similar lines of research, understood based on the keywords. To this end, the three-field graph integrates information regarding institutions (AU_UN), authors (AU), and keywords (DE, ID). It is observed that the gray flows between the first two columns demonstrate that many studies are the result of collaboration between different institutions through the support of authors (Figure 11a). Likewise, the flows between the central and right columns suggest, based on the keywords, that many authors share the same area of knowledge in different institutions, which can provide information feedback [76,77] and new insights. Although they share similar themes, such as soybean production, the studies may differ in purpose, focusing on performance and yield or deforestation. The variation depends on the spatial frontier adopted or the specific link in the production chain assessed. The overarching idea is to extend the boundaries and connections within the production chain, facilitating the integration of results obtained at each production stage. This can be achieved through projects that complement one another and establish links across the entire chain. The figure also illustrates collaborations among multiple authors, indicating a common research direction with similar objectives. Based on the figure, the suggestion is to broaden the thematic scope of studies by incorporating diverse perspectives and exploring additional research themes. Furthermore, it is possible to suggest potential thematic areas in which authors could collaborate to form public policies and propose the formation of multi-collaborative research groups between institutions, promoting research based on themes of common interest (Figure 11b). As an example, we can suggest the collaboration between de Faccio Carvalho, P. C.; Cherubin, M. R.; Cerri, C. E. P.; Cerri, C. C.; Crusciol, C. A. C.; Martins, A. P.; Hungary, M.; Franzluebbers, A. J.; and Anghinoni, I.; for the formation of public policies aimed at issues related to carbon sequestration or/and carbon emissions in agriculture and livestock.

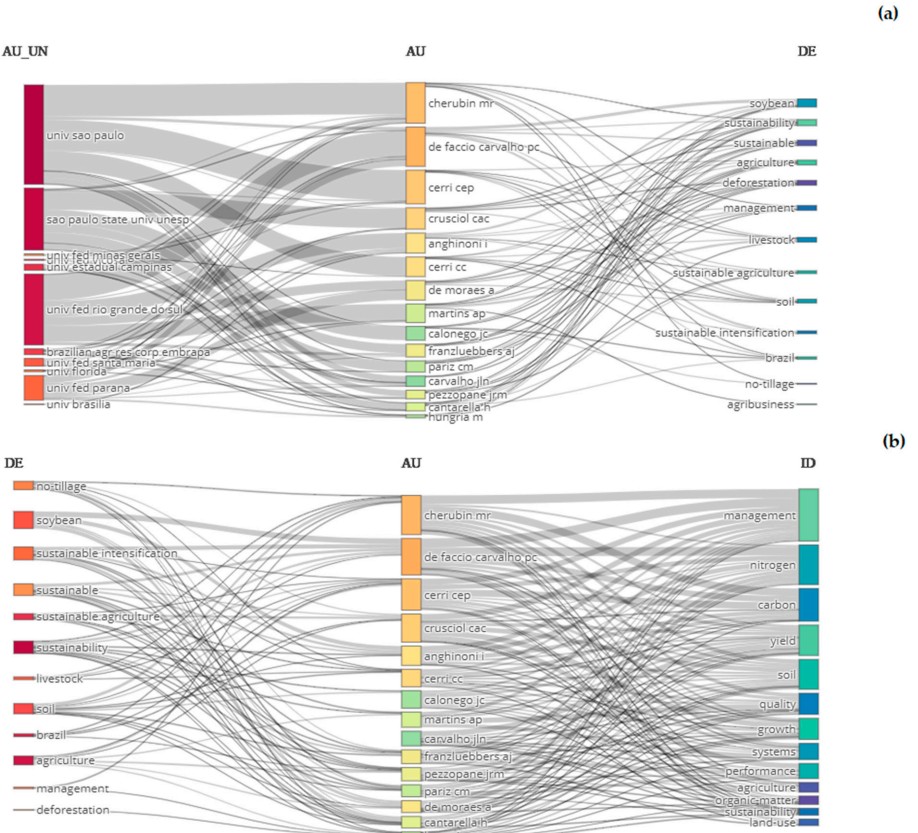

**Figure 11.** Three-field plot among institution (AU_UN), authors (AU), and author's keywords (DE; (**a**)); and among author's keywords and keyword Plus (ID; (**b**)). Thicker flows represent a greater relationship between produced documents and using keywords in documents.

*3.7. Thematic Evolution*

Figure 12 illustrates the evolution and direction of research topics over time. Starting from square and rectangular shapes, the flows (in gray) evolve from left to right, forming connections between keywords. Over time, there has been increasing use of the term "sustainability" as well as its branching into semantically similar terms, i.e., "sustainable agriculture" and "sustainable intensification." For instance, the Brazilian 'Semi-arid' region (Caatinga) has been extensively studied to enhance productivity through the adoption of sustainable practices in agricultural production [78–80].

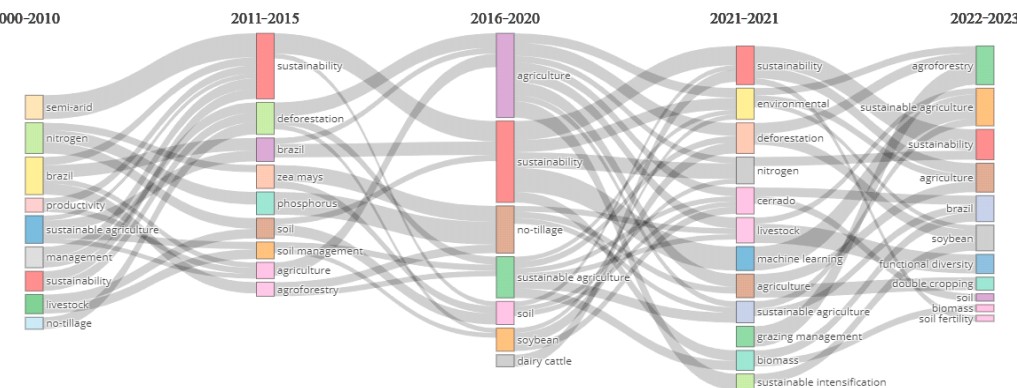

**Figure 12.** Thematic evolution of the author's keywords. Thicker streams represent a greater relationship between keywords. Larger shapes indicate keywords with a greater number of inbound and outbound streams.

It is worth highlighting the absence of terms such as "agrarian reform", "vulnerable population", "family farming", and "food security", among others. This fact indicates a possible sample bias for studies that include research more focused on modernization models in agricultural systems [81]. Terms such as "double-cropping", "biomass", "corn", and "soybean" can support this statement; however, there is also a possible evolution of research toward agricultural production concerned with the environment. This fact can be identified both from keywords that suggest more environmentally friendly production techniques (e.g., "agroforestry", "pasture management", and "no-tillage") and better management of potentially polluting agents (e.g., "nitrogen " and "phosphorus").

### 3.8. Future Vision of Brazilian Publications on Sustainability in Agriculture and Livestock

Figure 13 presents the vision for future publications using mapping analysis from the author's keywords (a) and the author's keywords on a thematic map (b). The clusters of the co-occurrence network are demonstrated as circles plotted graphically according to Callon's centrality and density classification [82]. The larger the size of the circle reflects, the greater the occurrence of the term. The *x*-axis shows the centrality of the network cluster, or the degree of interaction with other groups in the graph, and measures the importance of a study topic. The *y*-axis represents the density that measures the internal strength of a network of clusters and the growth of the theme [83–85].

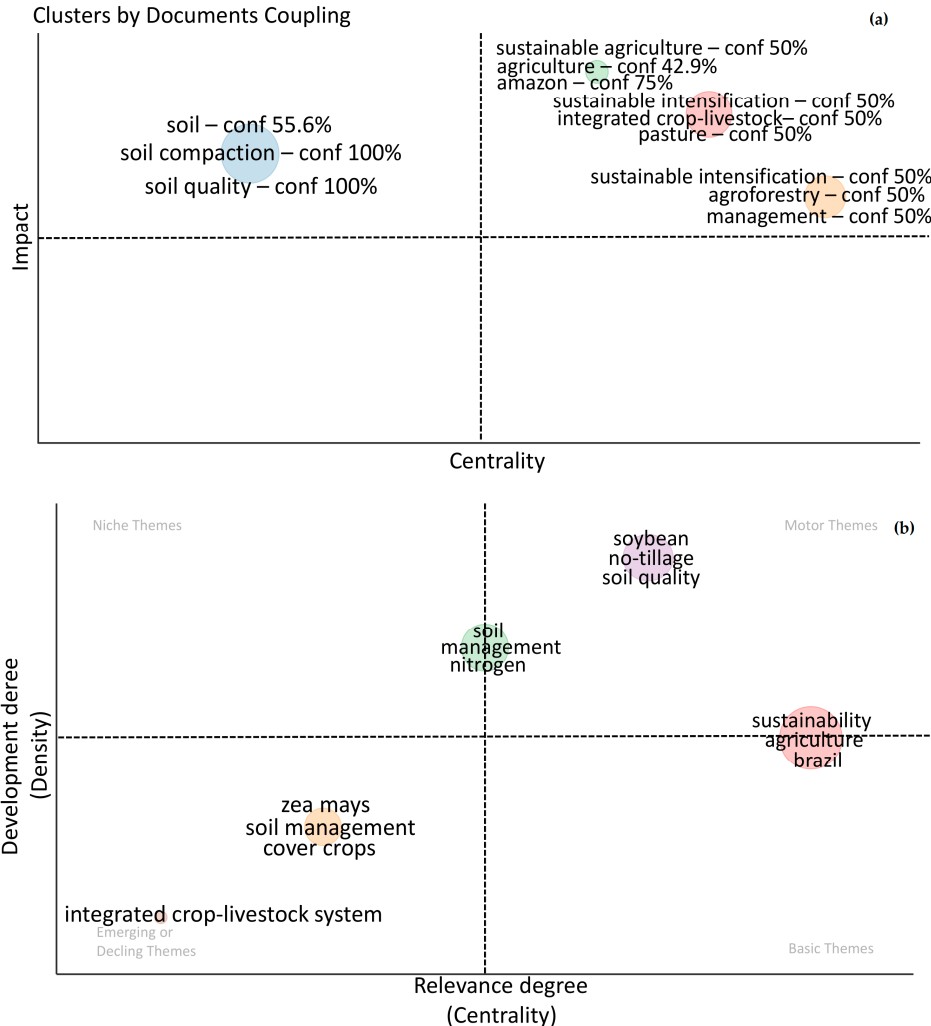

**Figure 13.** Visualization of (**a**) coupling cluster map considering the labeling of groups based on the author's keywords and (**b**) author keyword thematic map view.

According to the results, most terms found occupied the upper right quadrant (Figure 13a). Terms occupying the upper right quadrant are suggested as prominent themes characterized by high density and centrality; therefore, these themes need to be developed and are important to study in future research [86]. In this quadrant, the suggestion of studies related to "sustainable intensification", "crop–livestock integration", and "agroforestry" stood out. More specific themes are concentrated in the upper left quadrant; however, there is high development, as indicated by themes with high density and low centrality [86]. Topics related to soil (i.e., "soil quality" and "soil compaction") were highlighted in this quadrant.

Figure 13b demonstrates the formation of the quadrants, presenting them more intuitively. In this sense: (i) motor themes (top right quadrant), which shows the network of clusters with high density and centrality, position themes that are well-developed and crucial to structuring an area of knowledge; (ii) niche themes (top left quadrant), which presents themes with high density and low centrality, positioning themes of limited relevance; (iii) emerging or declining themes (lower left quadrant), which present themes with low density and centrality, implying minimally developed and marginal themes; and (iv) basic themes (lower right quadrant), with high centrality and low density, represent vital themes for the trans-disciplinarity of research [87]. Corroborating the results in Figure 13a, themes related to soil (i.e., "soil quality" and "no-tillage") present themselves as well-developed, as well as "soy". Themes such as "nitrogen" and "management", as well as "sustainability" and "agriculture", are themes presented on the border of the quadrant, referring to motor themes and niche and basic themes, respectively. This fact implies that such themes are on their way to becoming driving themes in the future with the increase in studies in the area of knowledge. Also, it can be inferred that "sustainability", "agriculture", and "Brazil" are themes of trans-disciplinary interest. Other themes such as "corn" (Zea Mays), "soil management", "cover crops", and "crop–livestock integration" were concentrated in the lower left quadrant (emerging or declining themes), which may indicate themes that are emerging from lines of research.

## 4. Conclusions

The study highlighted that, between 2000 and 2022, there was an exponential evolution in the Brazilian production of documents that correlate sustainability and agricultural production. Also, the leading role of universities such as USP, UNESP, UFRGS, and UFV, among others, was observed, playing an important role in articulating national and international partnerships for producing and disseminating scientific studies. Names such as Faccio Carvalho, P. C.; Cherubin, M. R.; Cerri, C. C.; Cerri, C. E. P.; and Crusciol, C. A. C., among others, were suggested as potential articulators and influencers for future research on the topic since they presented themselves as some of the most numerically productive authors. The keywords show that many authors share the same area of knowledge, which can provide information feedback and new insights for research and innovation. Also based on the keywords, the results suggested that the studies sampled were closely related to themes encompassing SDGs 1, 2, 6, 7, 12 and 13. Such evidence could support the proposition of forming multi-collaborative research groups between institutions, promoting research based on common interest themes, enabling the participatory elaboration of public policies as more sustainable paths for agricultural production.

The main documents cited in the sample literature encompass various features, from anthropogenically made soils and technologies for soil quality improvement (e.g., biochar fertilization) and their potential tools for sustainable agriculture (Lehmann et al. [37]). Additionally, they introduce a new plant trait database and discuss the importance of plant traits in understanding ecosystem functioning and their potential applications in various research fields (Kattge et al. [38]). The literature also presents the benefits of integrated crop–livestock systems for more sustainable agricultural production (De Moraes et al. [57]). Other cited documents highlight statistical procedures (R Core Team) and describe experimental conditions (Alvares et al. [66]).

Regarding thematic evolution, the results suggest a possible evolution of research towards agricultural production concerned with the environment. The results point to the increasing use of the term "sustainability" as well as its branching into semantically similar terms, i.e., "sustainable agriculture" and "sustainable intensification." Topics such as "sustainable intensification", "crop–livestock integration", and "agroforestry" were highlighted as being of important development in future research. However, although some were presented superficially, themes such as "agrarian reform", "vulnerable population", "family farming", and "food security", among others, were absent in the sample studied, indicating a possible bias for studies that consider research more focused on modernization models in agricultural systems. Since the accuracy of the results is closely linked to the search carried out using keywords, the study's main limitation is based on the low multidisciplinary nature of the search terms used being limited to WoS database. Also, while the search using terms in English covered documents published in other languages, we may have only retrieved a portion of the material produced on the subject. This could limit our understanding of the studies and introduce bias into the results and conclusions. Therefore, deciding not to use terms in other languages can also be a limitation. It is possible that the search strategy used in this study limited access to authors and lines of research from other areas of knowledge (i.e., social sciences) that would be as relevant to sustainability in agriculture as those presented. In addition, adding documents to the sample by expanding the search to new databases (i.e., Scopus and ScienceDirect) and including Portuguese, Spanish, and French terms could make the analysis more robust and comprehensive. Therefore, future studies must incorporate plural keywords in other languages and another scientific literature database that reaches other disciplines into their search strategy. This fact will provide a more appropriate search to contribute to expanding the multidisciplinary approach necessary for discussing proposals aimed at sustainability in the agricultural production chain as a whole.

The intricate relationship between agriculture and the Sustainable Development Goals (SDGs) underscores various challenges related to climate change, biodiversity loss, terrestrial ecosystem destruction, excessive consumption, and water pollution. As highlighted in this systematic review, addressing these complex issues requires a systematic and comprehensive approach. This review identifies key research groups, themes, and references, and integrates the rapidly increasing information across various scales of papers and periodicals. Such insights from dedicated research groups can offer valuable guidance for sustainable agriculture, focusing on critical factors like soil regulation, water consumption, no-tillage practices, livestock management, and other essential aspects. The future of agriculture is promising in the era of "sustainability", distant from the ideological filters, simplifications, and unilateral perspectives that undermine efforts to achieve sustainable development. Understanding and integrating all facets of agriculture, SDGs, and public policy, starting at the genetic level, necessitates more than intuition alone. Recognizing these challenges, policymakers should advocate for a holistic approach that combines science, public and private management, and stakeholder representatives in a collaborative and co-creative environment. Sustainability emerges from choices that consider the best possibilities to contribute to a pluralistic society grounded in egalitarian principles, promoting benefits for nature and respecting others. Science plays a pivotal role in describing and understanding the effects of the Brazilian agricultural sector, guiding decision-makers toward choices that align with sustainable development goals. This collaborative effort is essential for effective policymaking and the long-term well-being of society and the environment.

**Author Contributions:** Conceptualization, R.A.N. and V.T.R.; methodology, F.J.M.O., S.A.C. and R.A.N.; validation, R.A.N., F.P.R. and A.H.G.; formal analysis, V.T.R. and R.A.N.; investigation, R.A.N.; resources, R.A.N., F.J.M.O. and S.A.C.; data curation, R.A.N. and F.J.M.O.; writing—original draft preparation, R.A.N. and V.T.R.; writing—review and editing, R.A.N., S.A.C. and M.S.B.; visualization, R.A.N.; supervision, F.P.R. and A.H.G.; project administration, A.H.G. and F.P.R. All authors have read and agreed to the published version of the manuscript.

**Funding:** This research was funded by the Thematic Axes Program of the University of São Paulo (ProETUSP), grant number 972.

**Institutional Review Board Statement:** Not applicable.

**Informed Consent Statement:** Not applicable.

**Data Availability Statement:** The data presented in this study are available in the main text, where no new data were created.

**Acknowledgments:** This research was funded by Brazilian agencies: the Coordination of Superior Level Staff Improvement—CAPES, the National Council for Scientific and Technological Development—CNPq, and the São Paulo Research Foundation—FAPESP. R.A.N., F.J.M.O., S.A.C., M.S.B., A.H.G., and F.P.R thanks all the support provided by USP's rectory (PRPI—Pró-Reitoria de Pesquisa e Inovação da USP) from ProETUSP. SAC acknowledge the Coimbra Group Scholarship Programme for Young Professors and Researchers from Latin American Universities 2021.

**Conflicts of Interest:** The authors declare no conflicts of interest.

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
