# Peer review of "Sustainability and Brazilian Agricultural Production: A Bibliometric Analysis"

_sustainability, doi:10.3390/su16051833_

Round 1

Reviewer 1 Report

Comments and Suggestions for Authors

In this manuscript, the authors investigated relevant research articles about agriculture by Brazilian institutions. They found the use of sustainability” become more and the related papers grew rapidly. I think this work can be accept after the following minor comments:

1.     The authors may give the citation situation of the research articles to show their influence.

2.     Outlook part should be supplemented with the conclusion, please give your opinion about the limitation and future about the topic you discussed.

3.     More recent and related references should be supplemented.

4.     Maybe several tables can be added to give summary for the discussed parts.

5.     There are some language errors in the manuscript. The manuscript should be carefully checked.

Comments on the Quality of English Language

Minor editing of English language required.

Author Response

=> Responses to Reviewer #1:

We thank the referee for his/her thorough assessment of the manuscript. We now address the points raised by him/her.

Responses point-to-point

Referee #1: “The authors may give the citation situation of the research articles to show their influence.”

Response: Thank you for your valuable suggestion. We appreciate the opportunity to enhance the manuscript. In response to your request, we have included a new subsection, "3.4 Citation Assessment” (Lines: 439-487; p. 10-12), which delivers a comprehensive analysis of the citation situation for the research articles. This subsection not only presents citation indices but also includes information on works, influence, and the main themes discussed in the manuscripts. Please, check it.

Referee #1: “Outlook part should be supplemented with the conclusion, please give your opinion about the limitation and future about the topic you discussed.”

Response: Thank you for your suggestion. We appreciate your feedback, and in response to your comments, we have enhanced the Conclusion section by incorporating the future of agriculture, sustainability, and public policy (Lines 686-707; p.16). Additionally, we have addressed the limitations of the bibliometric analysis in subsection "2.1 Bibliometric Analysis” (Lines: 112-130; p.3) and Conlusion section (Lines: 672-676; p. 15).

Referee #1: “More recent and related references should be supplemented.”

Response: We have thoroughly revised our manuscript and incorporated additional references to enhance the comprehensiveness of our work. Additionally, a new subsection titled "3.4 Citation Assessment" (Lines: 440-487; p. 10) has been included, highlighting the most locally and globally cited documents/references. In the "4 Conclusions" section, we have reevaluated the five most significant works published since 2000, incorporating the highlighted references resulting from our analysis (Lines: 651-659; p. 15). The revised sections now reflect a more comprehensive and current overview of the relevant literature.

Referee #1: “Maybe several tables can be added to give summary for the discussed parts.”

Response: We believe that introducing additional tables is unnecessary, particularly with the supplementary information provided in the news table in subsection 3.4 "Citation assessment". All the analyses, along with the corresponding tables and plots for the reported results in the manuscript, have been explicitly detailed in the figure captions or the main text. Furthermore, to adhere to the specified length requirements established by Sustainability, we respectfully choose not to introduce any new tables.

Referee #1: “There are some language errors in the manuscript. The manuscript should be carefully checked.”

Response: We appreciate the comment and the manuscript was revised and changed properly in order to improve the language.

Reviewer 2 Report

Comments and Suggestions for Authors

Dear authors,

Congratulations for your work. The bibliometric analysis is exhaustive and it shows changes, patterns, connections and future areas of collaboration among a relevant group of Brazilian institutions and researchers

I have only a main concern regarding the sample you have used. You mention you used publications from WoS and in Eneglish and I wonder how relevant are publications in this field not indexed in WoS and /or in other languages like Portuguese and/or Spanish. I am  aware Brazil has a strong network of scientific journals in these languages and I wonder if leaving them out of the sample studied could be relevant. I mention this question because the conclusions you achieved, especially regarding the changes in keywords, or the absence of some themes may be challenged if there is not the same trend in those non English publications. I think some explanation on why you have excluded those publications is needed. even to mention that there are no relevant publications or the number is small

Regarding the paper I have some questions  

- The term "keywords plus" is used sometimes as it is and sometimes as "plus keywords", please review it

- Section 3, subsection 3.1 Line 183: You mention the average publication per year is 5 documents. Looking at the numbers in Figure 1 b) I guess you mean 5 documents per year in each source, is that correct? Probably it would be worth clarify this point

- In the same subsection you mention you have acquired 3139 documents comprising 22161 authors but in the aforementioned figure the number shown is 21380. Why is there a discrepancy ?

- In subsection 3.2 you mention the case of UNESP and teh use of two names. Three questions here, why do you don't add them in Figure 2.a showing its relevance and only mention the sum of documents? Is the university not unified in WoS as a single entrance? Why this duplicity only appears in this university and not in others?

- In the same subsection, but in line 252 you mention Embrapa is shown in red but it is not seen in Figure 3

- In section 3.3, when analysing authors, and especially in the figures you show, it would be interesting to know their affiliation, especially the most relevant. I understand it apperars in Figure 3 but it could help to follow the explanation. By the way, did you notice any change on affiliations of these authors? It would be interesting to know if any change has brought some connections among institutions.

- I would suggest a broader explanation on how to calculate H, G and M indexes. There are explanations on what they can show but probably it is worth to explain how they are calculated

 -I suggest to improve the explanation of Figure 10. It is not clear to reach to the suggestions you conlclude on possibel collaborations. The connection from an institution to an author may show already a collaboration that you suggest to form when it is already made. Moreover the connections made in Figure 10 b are not clear either

- I have also some problems in connecting the initial terms (2000-10) in Figure 11 to the second ones (2011-15). Why Semi arid is connected to Sustainability, for instance? 

In relation to figures:

- Figure 1 a: I would suggest to improve the graph by ading values in the axis to knwo which are the maximum values. In the x-axis I understand it is 2022 but in the y-axis it is not clear and this addition could help to undertand the values

- Figure 2 b:  What are the meaning of the colours? Sao Paulo University is clearly seen, but the other main universities cited in the text are not clearly shown 

- Figure 3: red and brown connections are not showing relevant names

- Figure 12: in the descirption there is no mention to keyword plus while in the text (row 409) they are described as used

Please consider these comments because I think it may help to improve the paper and make it more clear for readers

Author Response

=> Responses to Reviewer #2:

We appreciate the referee for their assessment of the manuscript and for stating, "Congratulations for your work". The referee also noted that "The bibliometric analysis is exhaustive and it shows changes, patterns, connections, and future areas of collaboration among a relevant group of Brazilian institutions and researchers". We have taken into consideration the points raised by the referee with the expectation of improving the manuscript.

Responses point-to-point

Referee #2: “I have only a main concern regarding the sample you have used. You mention you used publications from WoS and in Eneglish and I wonder how relevant are publications in this field not indexed in WoS and /or in other languages like Portuguese and/or Spanish. I am  aware Brazil has a strong network of scientific journals in these languages and I wonder if leaving them out of the sample studied could be relevant. I mention this question because the conclusions you achieved, especially regarding the changes in keywords, or the absence of some themes may be challenged if there is not the same trend in those non English publications. I think some explanation on why you have excluded those publications is needed. even to mention that there are no relevant publications or the number is small.”

Response: Thank you for the comment. In our analysis, we chose the Web of Science database because it captures metadata from key indexed journals, i.e., those with a careful process of editing and peer review, including those that are highly cited with a significant impact factor. In our database, we identified metadata related to four languages: Portuguese, French, Spanish, and English. Although the search was conducted in English, the WOS database can capture metadata from articles published in different languages as long as they have titles, abstracts, or keywords also written in English. Currently, national journals that publish scientific articles in local languages like those mentioned above also request that authors provide metadata in English. It is these metadata that are collected by the database used for bibliometric research. Furthermore, in the Brazilian Journal case, as mentioned by the referee, rigorous scientific journals are indexed by Scielo. WOS retrieves the metadata stored by Scielo.

To clarify this point, we have inserted a new paragraph in the "2.3 Data collection" section (Lines 158-174; p.4). Please check for it.

Referee #2: “The term "keywords plus" is used sometimes as it is and sometimes as "plus keywords", please review it”

Response: We appreciate the comment and the manuscript was revised and changed properly in order to improve the language.

Referee #2: “Section 3, subsection 3.1 Line 183: You mention the average publication per year is 5 documents. Looking at the numbers in Figure 1 b) I guess you mean 5 documents per year in each source, is that correct? Probably it would be worth clarify this point”

Response: Thank you for your question. The data mentioned in the previous version regarding the average publication per year being ~6 documents actually refers to the average age of the documents. This means that most documents were published in the last 6 years. The corrected note is as follows:

"The annual publication rate of documents was ~143, presenting an annual growth rate of ~17%. The average age of documents is ~6 years, indicating that most documents were published in the last 6 years". (Lines 238-243; p.5).

Referee #2: “In the same subsection you mention you have acquired 3139 documents comprising 22161 authors but in the aforementioned figure the number shown is 21380. Why is there a discrepancy ?”

Response: Thank you for bringing this to our attention. After carefully reviewing our analysis, we identified the correct number of authors as 21,380, as indicated in the second line of the subsection '3.1. Descriptive analysis of the selected sample' and reflected in Fig 1-b. We appreciate your diligence in ensuring the accuracy of our data.

Referee #2: “In subsection 3.2 you mention the case of UNESP and teh use of two names. Three questions here, why do you don't add them in Figure 2.a showing its relevance and only mention the sum of documents? Is the university not unified in WoS as a single entrance? Why this duplicity only appears in this university and not in others?”

Response: Thank you for bringing this to our attention. We have revisited and revised our analysis, incorporating Affiliation Name Disambiguation. During this process, we identified that Sao Paulo State University had three distinct names for the same institution (Sao Paulo State Univ Unesp, Sao Paulo State Univ, and Univ Estadual Paulista). In order to eliminate ambiguity, we have standardized the names to Sao Paulo State Univ Unesp. This standardization was applied consistently to all affiliations in the previously analyzed data, including the network depicted in Figure 2-b.

Moreover, we have updated the caption of Figure 2 to accurately.

“Distribution of papers by Brazilian institutions and collaboration networks of affiliations using bibliometric metadata from Web of Science (WoS). (a) Top 10 affiliations in article count; (b) Collaboration network among affiliations. Nodes represent institutions, and links depict collaboration between two affiliations. The University of São Paulo is highlighted as a major node due to its significant number of published articles. Other institutions are represented by nodes of size proportional to the number of documents published. Thicker links indicate the high intensity of collaboration between the institutions in relation to the publication of papers. The color of an item is determined by the cluster to which the item belongs” (Lines 319-326; p. 8).

Referee #2: “In the same subsection, but in line 252 you mention Embrapa is shown in red but it is not seen in Figure 3”

Response: Thank you for this point. We have revised the collaboration network, specifically labeling the clusters of research affiliations, which includes the Brazilian Agricultural Research Corporation (Embrapa). This adjustment aims to enhance the clarity of Figure 3, and we have updated the caption accordingly, as shown below

Author's co-occurrence network. Nodes represent authors, and links indicate collaboration between them for document productions. Larger nodes represent authors with a greater number of publications. The thickness of the links connecting authors signifies the intensity of collaboration, with thicker lines indicating a higher degree of cooperation between authors. The color of each node corresponds to the cluster to which the author belongs. Affiliation clusters are identified as follows: a) University of São Paulo (USP - blue); b) São Paulo State University (UNESP - red); c) Federal University of Rio Grande do Sul (UFRGS - green); and d) Brazilian Agricultural Research Corporation (Embrapa - gold). (Lines: 334-341; p. 8).

Referee #2: “In section 3.3, when analysing authors, and especially in the figures you show, it would be interesting to know their affiliation, especially the most relevant. I understand it apperars in Figure 3 but it could help to follow the explanation. By the way, did you notice any change on affiliations of these authors? It would be interesting to know if any change has brought some connections among institutions.”

Response: As the Reviewer #2 pointed out so well, the affiliation of each author could be find in the Figure 3. Also, the authors and its institution are demonstrated in the Figure 10-a (now Fig. 11-a and -b). Also, a brief explanation with Institution, theme and research lines were inserted Lines: 358-378; p. 8). In fact, it would be interesting to know if any change brought new connections among institutions. However, any author affiliation changes were noticed.

Referee #2: “I would suggest a broader explanation on how to calculate H, G and M indexes. There are explanations on what they can show but probably it is worth to explain how they are calculated”

Response: Thank you for your inquiry. To make clearer how the used indexes were claculated, the paragraph “The h-index is determined by sorting the author's publications in descending order of the number of citations each has received. The h-index is the highest number where the author has that many publications with at least that many citations each. Equation: If the author has "h" papers that have at least "h" citations each, but not more than "h+1" citations, then their h-index is "h." The m-index is another variant that considers both the number of publications and their citations. It is calculated by dividing the sum of the square roots of the number of citations of each paper by the square root of the total number of publications. Equation: M = Σ√c / √N, where Σ√c is the sum of the square roots of the number of citations, and √N is the square root of the total number of publications.  The g-index considers not only the number of citations but also the "weight" of each publication. It is calculated by dividing the sum of the square roots of the number of citations of each paper by the sum of the square roots of the natural numbers. Equation: G = (Σ√c) / (Σ√n), where Σ√c is the sum of the square roots of the number of citations, and Σ√n is the sum of the square roots of the natural numbers (publication order)” was included (Lines 205-218; p.5).

Referee #2: “I suggest to improve the explanation of Figure 10. It is not clear to reach to the suggestions you conlclude on possibel collaborations. The connection from an institution to an author may show already a collaboration that you suggest to form when it is already made. Moreover the connections made in Figure 10 b are not clear either”

Response: We appreciate your feedback and acknowledge the need for clarity in Figure 10. To enhance comprehension, we have included the following sentence:

Although they share similar themes, such as soybean production, the studies may differ in purpose, focusing on aspects like performance and yield or deforestation. The variation is dependent on the spatial frontier adopted or the specific link in the production chain assessed. The overarching idea is to extend the boundaries and connections within the production chain, facilitating the integration of results obtained at each production stage. This can be achieved through projects that complement one another and establish links across the entire chain. The figure also illustrates collaborations among multiple authors, indicating a common research direction with similar objectives. Based on the figure, the suggestion is to broaden the thematic scope of studies by incorporating diverse perspectives and exploring additional research themes” (Lines 546-555; p. 13).

Referee #2: “I have also some problems in connecting the initial terms (2000-10) in Figure 11 to the second ones (2011-15). Why Semi arid is connected to Sustainability, for instance?”

Response: Thank you for this point. Objetiving clarify this point, we have insert a sentence

For instance, the Brazilian 'Semi-arid' region (Caatinga) has been extensively studied with the aim of enhancing productivity through the adoption of sustainable practices in agricultural production” (Lines: 576-578; p. 13).

Referee #2: “Figure 1 a: I would suggest to improve the graph by ading values in the axis to knwo which are the maximum values. In the x-axis I understand it is 2022 but in the y-axis it is not clear and this addition could help to undertand the values”

Response: Thank you for your suggestion. We have updated Figure 1a to include maximum values on the y-axis. Please verify the revised version.

Referee #2: “Figure 2 b:  What are the meaning of the colours? Sao Paulo University is clearly seen, but the other main universities cited in the text are not clearly shown”

Response: We have revised the figure caption to encompass all necessary information, including details about nodes, links, their respective sizes, and colors. The updated caption is provided below:

Distribution of papers by Brazilian institutions and collaboration networks of affiliations using bibliometric metadata from Web of Science (WoS). (a) Top 10 affiliations in article count; (b) Collaboration network among affiliations. Nodes represent institutions, and links depict collaboration between two affiliations. The University of São Paulo is highlighted as a major node due to its significant number of published articles. Other institutions are represented by nodes of size proportional to the number of documents published. Thicker links indicate the high intensity of collaboration between the institutions in relation to the publication of papers. The color of an item is determined by the cluster to which the item belongs”. (Lines: 319-326)

Referee #2: “Figure 3: red and brown connections are not showing relevant names”

Response: Thank you for the point. We chose to present the entire collaboration network. The significant names can be identified by the size of the nodes and thickness of the edges, as indicated in the figure legend, as shown below:

Author's co-occurrence network. Nodes represent authors, and links indicate collaboration between them for document productions. Larger nodes represent authors with a greater number of publications. The thickness of the links connecting authors signifies the intensity of collaboration, with thicker lines indicating a higher degree of cooperation between authors. The color of each node corresponds to the cluster to which the author belongs. Affiliation clusters are identified as follows: a) University of São Paulo (USP - blue); b) São Paulo State University (UNESP - red); c) Federal University of Rio Grande do Sul (UFRGS - green); and d) Brazilian Agricultural Research Corporation (Embrapa - gold).” (Lines 334-341; p. 8)

Referee #2: “Figure 12: in the descirption there is no mention to keyword plus while in the text (row 409) they are described as used”

Response: Thank you for your observation. The manuscript was reviewed to correct any writing errors.

Reviewer 3 Report

Comments and Suggestions for Authors

Highlights:

1. We think that they are general concepts and results. It is not the highlight of this research. We recommend to compare the difference with other articles, and evaluate the contribution, importance and creativity into the highlights.

Introduction

1. Line 39, the symbol ")" is missing here.

2. Line 39-40, regarding “although questions still remain about its current reality regarding such responsibility.”, we do not understand the meaning and suggest an explanation.

3. The introduction covers some references, however does not introduce the scientific problem. How will this study contribute beyond the current literature?

Materials and Methods

1. Line 135, 149, regarding "2.2. Data collection", the order of subtitles is wrong, please correct it.

2. Why not use other software, such as VOSviewer, CiteSpace, etc. How to confirm the effectiveness of software execution?

3. Why not use other databases, such as Scopus, Google Scholar, etc. How to confirm that the database is sufficiently covered?

4. Are some publications in local languages? How are non-English documents handled?

5. Line 162, it is recommended to state these indicators.

Results and Discussion

1. Line 182, Regarding "comprising 22,161 authors", our question is whether there are so many authors (number of different authors), or the frequency of authors (including duplicate authors)?

2. Some Figures are not displayed clearly. It is recommended to improve and increase the resolution. For example, Figure 2b, Figure 10, Figure 11, etc.

3. Line 241, 248, regarding "Thicker flows", it is recommended to explain clearly.

4. Line 285-314, the authors used three Indexes (H index, G index and M index) in the article. We do not understand the reasons for the differences between the three. Why use these three Indexes?

Author Response

=> Responses to Reviewer #3:

We thank the referee for his/her thorough assessment of the manuscript. We address the points raised by him/her expecting to improve the manuscript.

Responses point-to-point

Referee #3: “Concerning Highlights, we think that they are general concepts and results. It is not the highlight of this research. We recommend to compare the difference with other articles, and evaluate the contribution, importance and creativity into the highlights”

Response: The authors would like to thank you for your suggestion. We suggested a new highlight and made corrections to make the contribution, importance and creativity of this study clearer.

Referee #3: “Line 39, the symbol ")" is missing here.”

Response: Thank you for your comment. The highlighted term has been fixed.

Referee #3: “Line 39-40, regarding “although questions still remain about its current reality regarding such responsibility.”, we do not understand the meaning and suggest an explanation.”

Response: The comment is related to national political divergences about the role of Brazilian agriculture in its current state and its implications for combating hunger and food security. Since the sentence is out of the context of the article, it has been removed to make the text clearer.

Referee #3: “The introduction covers some references, however does not introduce the scientific problem. How will this study contribute beyond the current literature?”

Response: Thank you for your observation and suggestion. At the end of Section 1. Introduction, five research questions were formulated to guide toward a broader objective: identifying key researchers, their respective works, topics, and collaboration networks focusing on the sustainability of Brazilian agricultural production. As a contribution to the current literature, we suggest a path for the formation of multidisciplinary research groups dedicated to solving current issues aligned with sustainable development goals.

Naturally, the selection of references is inevitably influenced by the authors' preferences and scientific tradition. Therefore, most of the references included in our manuscript report significant results related to the scope. The remaining references are intended to support and complement basic bibliometric analysis and systematic review, etc. We acknowledge that seminal documents related to the SDGs relevant to our work might be lacking. We are open to suggestions that enhance our references.

Referee #3: “Line 135, 149, regarding "2.2. Data collection", the order of subtitles is wrong, please correct it.”

Response: Thank you for your comment. We have corrected the ordering of subsections.

Referee #3: “Why not use other software, such as VOSviewer, CiteSpace, etc. How to confirm the effectiveness of software execution?”

Response: Thank you for your question. In our bibliometric analysis, we employed a combination of tools to ensure a comprehensive examination of the literature. Specifically, we utilized Bibliometrix, a robust R-based package, along with the Shiny app (biblioshiny), to conduct the bibliometric analysis. Additionally, we used VOSviewer and Mathematica for generating various analyses and plots to validate and cross-verify the results.

By leveraging multiple software tools, each with its strengths and capabilities, we aimed to enhance the reliability and effectiveness of our analysis.  The combination of Bibliometrix, biblioshiny, VOSviewer, and Mathematica facilitated a multi-faceted exploration of the data, contributing to the overall quality and validity of our bibliometric assessment.

We complement the information concerning computational tools employed in our analysis by inserting the following sentence into the first paragraph of the section:

In addition to Biblioshiny, we integrated VOSviewer and Mathematica into our analysis, enhancing the visualization and validation of our results. VOSviewer, renowned for its network analysis capabilities, allowed us to delve into collaboration patterns and thematic clusters within the literature. Mathematica, with its versatile numerical tools, complemented our approach by providing additional insights and facilitating a comprehensive examination of the bibliometric landscape. Together, these software tools were pivotal components of our analysis, ensuring a robust and multifaceted exploration of the bibliometric data” (Lines: 192-199; p. 4).

Referee #3: “Why not use other databases, such as Scopus, Google Scholar, etc. How to confirm that the database is sufficiently covered?”

Response: Thank you for your question. To make clear the choice by the used database, the paragraph: “The choice to use Web of Science as our primary database was influenced by several factors. Web of Science is a widely recognized and extensively used database, particularly in the academic and research community. It is known for its comprehensive coverage of peer-reviewed journals across various disciplines. The scope of this research, focusing on sustainability, agriculture, and related fields, aligns well with the content indexed in Web of Science, showing a robust platform for bibliometric analysis. While Scopus and Google Scholar are also valuable resources, Web of Science has been integrated into the Scielo platform since 2014, where a significant portion of peer-reviewed journals in multiple languages is indexed. Moreover, a recent study showed that despite evidence that Google Scholar significantly retrieves more citations than the WoS Core Collection and Scopus in all thematic areas, all citations found by WoS (95%) and Scopus (92%) were also found by Google Scholar. However, not all documents indexed by Google Scholar undergo a rigorous process of peer review. For these reasons, we chose the Web of Science database to guide our bibliometric analysis” was inserted (Lines: 158-171; p. 4).

Referee #3: “Are some publications in local languages? How are non-English documents handled?”

Response: Thank you for the comment. In our analysis, we chose the Web of Science database because it captures metadata from key indexed journals, i.e., those with a careful process of editing and peer review, including those that are highly cited with a significant impact factor. In our database, we identified metadata related to four languages: Portuguese, French, Spanish, and English. Although the search was conducted in English, the WOS database can capture metadata from articles published in different languages as long as they have titles, abstracts, or keywords also written in English. Currently, national journals that publish scientific articles in local languages like those mentioned above also request that authors provide metadata in English. It is these metadata that are collected by the database used for bibliometric research. Furthermore, in the Brazilian Journal case, as mentioned by the referee, rigorous scientific journals are indexed by Scielo. WOS retrieves the metadata stored by Scielo.

To clarify this point, we have inserted a new paragraph in the "2.3 Data collection" section (Lines: 158-171; p. 4). Please check for it.

Referee #3: “Line 162, it is recommended to state these indicators”

Referee #3:  “Line 285-314, the authors used three Indexes (H index, G index and M index) in the article. We do not understand the reasons for the differences between the three. Why use these three Indexes?”

Response: Thank you for your inquiry. We grouped the two questions above because they refer to the same nature of the question. In biblioshiny, a package of bibliometrix in R, the calculation of the H index, G index, and M index follows these formulations:

To make clearer how the used indexes were claculated, the paragraph “The h-index is determined by sorting the author's publications in descending order of the number of citations each has received. The h-index is the highest number where the author has that many publications with at least that many citations each. Equation: If the author has "h" papers that have at least "h" citations each, but not more than "h+1" citations, then their h-index is "h." The m-index is another variant that considers both the number of publications and their citations. It is calculated by dividing the sum of the square roots of the number of citations of each paper by the square root of the total number of publications. Equation: M = Σ√c / √N, where Σ√c is the sum of the square roots of the number of citations, and √N is the square root of the total number of publications.  The g-index considers not only the number of citations but also the "weight" of each publication. It is calculated by dividing the sum of the square roots of the number of citations of each paper by the sum of the square roots of the natural numbers. Equation: G = (Σ√c) / (Σ√n), where Σ√c is the sum of the square roots of the number of citations, and Σ√n is the sum of the square roots of the natural numbers (publication order)” was included (Lines 205-218; p.5).

Referee #3: “Line 182, Regarding "comprising 22,161 authors", our question is whether there are so many authors (number of different authors), or the frequency of authors (including duplicate authors)?”

Response: The point highlighted by the referee pertains to the frequency of authors. In our analysis, we identified a total of 21,380 authors contributing to 3,139 documents. It's noteworthy that an author may be counted multiple times in the total count of authors in the documents. For a more detailed explanation, we have modified the introductory paragraph of Section 3.1 as follows:

"Following the methodology, the study identified a total of 3,139 documents, comprising 21,380 authors distributed among all documents (Figure 1-b). The annual publication rate of documents was ~143, presenting an annual growth rate of ~17%. The average age of documents is ~6 years, indicating that most documents were published in the last 6 years. It is noted that interest in the topic studied gained prominence from 2007 onwards, with exponential growth since then, with its peak being observed in 2021 (n=546 publications), representing ~17% of the sample." (Lines 237-243; p. 5)

Referee #3: “Some Figures are not displayed clearly. It is recommended to improve and increase the resolution. For example, Figure 2b, Figure 10, Figure 11, etc”

Response: We have provided a new figure with better quality.

Referee #3: “Line 241, 248, regarding "Thicker flows", it is recommended to explain clearly.”

Response: Thank you for your feedback. To provide a clearer explanation, we have revised the caption of the figure as follows:

Author's co-occurrence network. Nodes represent authors, and links indicate collaboration between them for document production. Larger nodes represent authors with a greater number of publications. The thickness of the lines connecting authors signifies the intensity of collaboration, with thicker lines indicating a higher degree of cooperation between authors. The color of each node corresponds to the cluster to which the author belongs. Affiliation clusters are identified as follows: a) University of São Paulo (USP - blue); b) São Paulo State University (UNESP - red); c) Federal University of Rio Grande do Sul (UFRGS - green); and d) Brazilian Agricultural Research Corporation (Embrapa - gold)”. (Lines 334-341; p.8).

Reviewer 4 Report

Comments and Suggestions for Authors

Dear Editor

thank you for having me present to evaluate this work. It is a very interesting paper and related to the journal.

However, there are some changes that must be made and also include some disutions in the document, so that it can be accepted.

My comments are in the attached file.

Regards

Reviewer

Author Response

=> Responses to Reviewer #4:

We thank the referee for his/her thorough assessment of the manuscript. We address the points raised by him/her expecting to improve the manuscript.

Responses point-to-point

Referee #4: “Concerning section 2.1, please expand the paragraph, including key aspects, advantages and disadvantages of bibliometric analysis”

Response: Thank you for your suggestion. We appreciate your feedback and have expanded the paragraph in Section 2.1, providing more details on key aspects, advantages, and disadvantages of bibliometric analysis. The revised text can be found in the first paragraph of Section 2.1. Bibliometric Analysis and is also presented below

Bibliometric analysis is a widely utilized method for quantitatively evaluating scientific output and trends within a particular research field. Key aspects of bibliometric analysis include the examination of publication patterns, citation networks, collaboration structures, and identification of influential works, authors or institutions for the development of academic studies correlating different areas of knowledge [13]. This method offers several advantages, such as providing a systematic and objective way to assess the impact of research, uncovering emerging trends, and aiding in decision-making for future research directions. Additionally, bibliometric techniques contribute to the transparency and reproducibility of the review process. However, like any methodology, bibliometric analysis comes with its set of limitations. One notable limitation is its reliance on scientific databases, which may not be exclusively designed for bibliometric purposes and can introduce errors that require careful handling. The quantitative nature of bibliometrics can also pose challenges when transitioning to qualitative insights, making it crucial to supplement findings with content analysis for a more comprehensive understanding. Furthermore, while bibliometric studies offer valuable short-term insights, making ambitious claims about the long-term impact of a research field may be challenging. Despite these limitations, bibliometric analysis remains a valuable tool for navigating the complex landscape of scientific knowledge” (Lines 113-130; p. 3).

Referee #4: “In the last paragraph of Section 2.3, please delete year 2018 in reference [13]”

Response: Thank you. We have reviewed and corrected the paper appropriately to eliminate the formatting mistakes.

Referee #4: “In the first paragraph of Section 3.2, please include why these universities are leading the way in research”

Response: Thank you for your suggestion. To make the motivation of leading of the universities clearer, the paragraphs: “The prominence of the institutions listed in this study can be attributed to two factors: (i) number of research programs with high-quality scores ranging; and (ii) the available financial support. Regarding the scores ranging, periodically, the research programs associated with Brazilian institutions undergo governmental quality evaluations. In these evaluations, five criteria are considered: program proposal, teaching staff, student body, theses and dissertations, and intellectual production, as well as social insertion. Ultimately, these programs receive scores ranging from 1 to 7. A score of 5 indicates that the course is considered 'very good,' while scores of 6 and 7 are awarded to programs of excellence at the national and international levels, respectively. Among the 190 Brazilian research programs linked to the Agricultural Sciences field with scores between 5 and 7, approximately 57% (108 research programs) are affiliated with the top 10 institutions. Within this group, approximately 12% (23 research programs) are associated with USP, which boasts the highest number of research programs with international excellence (7 research programs). Most of these programs have over 50 years of experience, suggesting a high level of maturity.

In other hand, the financial support is closely related to these scores ranging and its maintenance, for instance, specific financial resource programs to support academic excellence programs (Programa de Excelência Acadêmica (Proex/Capes/MEC)). According to the National Council for Scientific and Technological Development database (CNPq) , other crucial federal research support agency, approximately 1,350 research institutions received financial backing for research development between 2002 and 2022 (including research grants, financial project support, etc.). Among these, the top 10 institutions garnered 48% (1.7 billion) of the total financial resources available. UFV, USP, UNESP, and UFRGS collectively received 26% (R$ 941.5 million) of the overall financial resources. It's noteworthy that this figure pertains specifically to the financial support allocated for Agricultural Sciences field.  Thus, it is possible to suggest that the prominence of those institutions has been related to a higher number of research programs that present high-quality score ranging” were inserted (Lines: 288-315; p. 7).

Referee #4: “Please, improve the Picture 2”

Response: Thank you for your suggestion. We improve all figures.

Referee #4: “In the second paragraph of Section 3.3, I consider that it is not appropriate to use acronyms for the name of the authors, please handle these names as is done in other bibliometric studies”

Response: Thank you for your suggestion. We have changed the format of presenting authors, following the traditional literature style, with the last name extended and the first and middle names abbreviated. Please, check it.

Referee #4: “Please, improve the Figure 4b”

Response: We have provided a new figure with better quality.

Referee #4: “Please, include more information on the 5 most relevant authors, institutional affiliations, themes or relevant topics they work on”

Response: We appreciate your feedback, and in response to your comments, we have included more detailed information about the 5 most relevant authors, including their institutional affiliations and the themes or relevant topics they work on. This information can be found in the newly added second paragraph of section 3.3 Authors. The inserted text can also be seen below:

Starting from the number of documents produced per author, the results suggest Faccio Carvalho, P. C.; Cherubin, M. R.; Cerri, C. E. P; Crusciol C. A. C.; and Cerri, C. C. as the five most numerically productive authors. Faccio Carvalho, P. C., has been a professor at the Department of Forage Plants and Agrometeorology at UFRGS since 1997, focusing on research and extension in the soil-plant-animal relationship. His studies encompass natural environments, cultivated lands, and integrated pasture grazing. Cherubin, M. R., is a professor at the Department of Soil Science at ESALQ-USP. His studies center around quantifying and comprehending the impacts of land use, emphasizing the management and health of soil practices, carbon dynamics, and the provision of ecosystem services in both natural and agricultural ecosystems. Cerri, C. E. P., is a professor at the Department of Soil Science at ESALQ-USP, with a focus on themes related to soil organic matter, global warming, climate changes, agriculture, and carbon credits, mathematical models, geostatistics, and geoprocessing. Cerri, C. C., has been a professor at the Center of Nuclear Energy in Agriculture (CENA/USP) at USP since 1975. He specializes in researching soil organic matter dynamics in natural ecosystems and those modified by agricultural practices, livestock, and reforestation. His studies specifically address greenhouse gas emissions, climate changes, and land-use changes associated with agriculture and livestock production practices. Crusciol, C. A. C., is a professor at the Faculty of Agricultural Science at UNESP (FCA/UNESP). His expertise is concentrated on agricultural production systems, soil fertility management, plant nutrition, and applied vegetal physiology” (Lines: 358-378; p. 8).

Referee #4: “In conclusion section, I consider need include a discussion above the main five research works”

Response: We appreciate the opportunity to enhance the manuscript. In response to your request, we have included a discussion above the main five research works in the second paragraph of the 4 Conclusions section. The added text can also be seen below:

The main documents cited in the sample literature encompass various features, from anthropogenically-made soils and technologies for soil quality improvement (e.g., biochar fertilization) and their potential tools for sustainable agriculture (Lehmann 2003). Additionally, they introduce a new plant trait database and discuss the importance of plant traits in understanding ecosystem functioning and their potential applications in various research fields (Kattge 2020). The literature also presents the benefits of integrated crop-livestock systems for more sustainable agricultural production (De Moraes 2014). Other cited documents highlight statistical procedures (R Core Team) and describe experimental conditions (Alvares 2014)” (Lines: 651-659; p. 15)

Round 2

Reviewer 3 Report

Comments and Suggestions for Authors

In the revised version, we have seen the author's efforts to explain and respond to reviewer comments, and make a meaningful correction. For our concerns, we can see that most of them have a new addition and improvement.

Reviewer 4 Report

Comments and Suggestions for Authors

Dear 

Editor, the authors have included the suggestions made by me.

Regards